# Collective chiroptical activity through the interplay of excitonic and charge-transfer effects in localized plasmonic fields

Huacheng Li[1,2,3,13], Xin Xu[1,2,3,13], Rongcheng Guan[1,2,3], Artur Movsesyan [4,5], Zhenni Lu[1,6], Qiliang Xu[1,6], Ziyun Jiang[1,2,3], Yurong Yang[1,2,3], Majid Khan[1,2,3], Jin Wen [1,3], Hongwei Wu[1,6], Santiago de la Moya[7], Gil Markovich[8], Huatian Hu[9], Zhiming Wang[4], Qiang Guo [10], Tao Yi [1,6], Alexander O. Govorov [4,5] ✉, Zhiyong Tang [11,12] & Xiang Lan [1,2,3] ✉

The collective light-matter interaction of chiral supramolecular aggregates or molecular ensembles with confined light fields remains a mystery beyond the current theoretical description. Here, we programmably and accurately build models of chiral plasmonic complexes, aiming to uncover the entangled effects of excitonic correlations, intra- and intermolecular charge transfer, and localized surface plasmon resonances. The intricate interplay of multiple chirality origins has proven to be strongly dependent on the site-specificity of chiral molecules on plasmonic nanoparticle surfaces spanning the nanometer to sub-nanometer scale. This dependence is manifested as a distinct circular dichroism response that varies in spectral asymmetry/splitting, signal intensity, and internal ratio of intensity. The inhomogeneity of the surface-localized plasmonic field is revealed to affect excitonic and charge-transfer mixed intermolecular couplings, which are inherent to chirality generation and amplification. Our findings contribute to the development of hybrid classical-quantum theoretical frameworks and the harnessing of spin-charge transport for emergent applications.

Molecular excitations interacting with confined light fields are expected to yield intriguing photophysical and/or photochemical phenomena. Recent studies reported vibrational modification of chemical reactivity[1], substantial enhancement of intersystem crossing[2], and other effects[3], which were observed in the optical cavity mode[4,5].

Despite their lossy nature, plasmonic nanoparticles (cavities) have been adopted to construct nanoparticle-molecular hybrid complexes due to the small mode volume of surface-concentrated optical fields. Chiral plasmonic hybrid complexes have demonstrated a collection of desirable chiral optical effects, such as plasmon-induced chirality

[1]State Key Laboratory for Modification of Chemical Fibers and Polymer Materials, Donghua University, 201620 Shanghai, China. [2]Center for Advanced Low-dimension Materials, Donghua University, 201620 Shanghai, China. [3]College of Materials Science and Engineering, Donghua University, 201620 Shanghai, China. [4]Institute of Fundamental and Frontier Sciences, University of Electronic Science and Technology of China, 610054 Chengdu, China. [5]Department of Physics and Astronomy and Nanoscale and Quantum Phenomena Institute, Ohio University, Athens, OH 45701, USA. [6]College of Chemistry and Chemical Engineering, Donghua University, 201620 Shanghai, China. [7]Departamento de Química Orgánica, Facultad de Ciencias Químicas, Universidad Complutense de Madrid, Ciudad Universitaria s/n, 28040 Madrid, Spain. [8]School of Chemistry, Raymond and Beverly Sackler Faculty of Exact Sciences, Tel Aviv University, Tel Aviv 69978, Israel. [9]Center for Biomolecular Nanotechnologies, Istituto Italiano di Tecnologia, Via Barsanti 14, Arnesano 73010 LE, Italy. [10]State Key Laboratory of Protein and Plant Gene Research, Center for Life Sciences, Academy for Advanced Interdisciplinary Studies, School of Life Sciences, Peking University, 100871 Beijing, China. [11]CAS Key Laboratory of Nanosystem and Hierarchical Fabrication & CAS Center for Excellence in Nanoscience, National Center for Nanoscience and Technology, 100190 Beijing, China. [12]University of Chinese Academy of Sciences, 100049 Beijing, China. [13]These authors contributed equally: Huacheng Li, Xin Xu. ✉e-mail: govorov@ohio.edu; xlan@dhu.edu.cn

transfer between molecules[6], from molecules to nanoparticles[7], or vice versa[8], molecular chirality amplification[9], and others[10–18]. These studies of chiral light–matter interactions in the confined light field have far-reaching implications for enantiomer discrimination, clinical theranostics[19], quantum information[20], etc. Nevertheless, clarifying the collective coupling of chiral supramolecular aggregates or molecular ensembles with localized surface plasmons has become a challenge.

To analyze the chiral plasmonic complexes, chiral molecules were theoretically treated as single, isolated two-level systems[21], a typical descriptive model of Frenkel excitons where the excited electronic wavefunction is localized on individual molecules[22,23]. The excitonic couplings in the molecular ensembles were not taken into account. However, recent experimental results of supramolecular chiral plasmonic complexes[24,25] demonstrated that coherent excitonic couplings can substantially reshape the chiral optical characteristics of the complexes. Such observations revisit the neglected role of exciton delocalization and coherence in the chiral molecular ensembles of the complexes. In addition to long-range coherent interactions, strong coupling due to short-range interactions, i.e., orbital overlap and hybridization[26], can occur in chiral molecular ensembles, leading to intermolecular charge transfer (CT) or exchange[27]. However, little is known about the mixing of Frenkel and CT states in chiral plasmonic complexes, although inherent electronic couplings and spin-charge transport[28] are fundamental to enantioselective catalysis[29], chiral optoelectronics[30], chiral quantum optics[31], etc.

Intra- and intermolecular CT processes are widespread in synthetic supramolecular assemblies of π-conjugated molecules, such as perylene diimide[32] and naphthalene diimide[33], as well as in biological systems, such as photosynthetic light-harvesting complexes[34] and others[35,36]. It was recently stressed that the conventional Kasha theory must be expanded by incorporating Frenkel-CT mixing for analyzing CT-involved, strongly coupled molecular ensembles[37]. Nonetheless, the intricate interplay of Frenkel excitonic couplings, intra- and intermolecular CT, and localized surface plasmons imposes great theoretical challenges, undoubtedly exceeding the current framework. Thus, to study the underlying mechanisms and pursuit of unveiled chiral phenomena, experimental models with controllable and quantitative parameters are needed. Accurate control of combined factors, such as energy level alignment between molecules and nanoparticles[38], molecular stacking on nanoparticle surfaces and spatial separation, and highly ordered arrangement of nanoparticles, is desired.

In this study, we report the use of DNA-programmable addressability[39,40] to construct chiral plasmonic complexes with intra- and intermolecular CT characteristics at different hierarchical levels, namely, the single-particle and superstructure levels. We realized the monolayer organization of chiral chromophores on nanoparticle surfaces with site-specificity at the nanometer to sub-nanometer scale. Circular dichroism (CD) spectroscopic measurements revealed a strong dependence of the spectral lineshape asymmetry/splitting, signal intensity, and internal ratio of intensity on the site-specificity of the chiral chromophores. Intermolecular CT was clarified via DNA templating of molecular dimer aggregates, ultrafast transient absorption spectroscopy, and quantum chemical calculations. Overall, it was found that Frenkel excitonic correlations and intra/intermolecular CT significantly affect the chirality generation and amplification of chiral molecular ensembles. By adjusting the nanoparticle dimensions and making monodisperse superstructures of nanoparticles with controllable positioning of chiral chromophores, the influence of localized plasmonic field was studied. CD measurements and electromagnetic simulations indicated that the varying CD signal intensities were strongly associated with the spatial inhomogeneity of the local electric field distribution. In particular, in superstructures, the core particle works as an "amplifying antenna"[41,42], enhancing the local electric field on the satellite–molecule complexes and, therefore, amplifying the collective chiroptical responses in a cascade-like fashion.

## Results

### Site-specificity and optical properties of hybrid complexes

For the first time, we synthesized site-specific conjugates of a DNA strand and a chiral BODIPY chromophore (BINOL-based O-BODIPY[43,44], denoted as cBDP, Supplementary Figs. 2–16, Supplementary Data 1) to enable monolayer organization of cBDP molecular ensembles on the surface of plasmonic Au/Ag nanoparticles (AuNPs/AgNPs, Fig. 1a, b). The distance of cBDP to the nanoparticle surface ranged from 14 nucleotides to 7 and 2 nucleotides (Au/Ag complexes with cBDP are denoted as 14nt, 7nt, and 2nt complexes, respectively). According to the estimation of 0.34 nm bp$^{-1}$ in the DNA double helix and considering the linker's flexibility, the distance can reach up to <1 nm scale for 2nt complexes. The cBDP–DNA conjugates exhibited almost the same absorption and CD spectra lineshape as those of pure cBDP molecules (Supplementary Fig. 14), demonstrating no observable influence of CT interaction[45] or off-resonant excitonic coupling between cBDP and neighboring DNA nucleotides[46].

The cBDP–nanoparticle hybrid complexes composed of cBDP (S) and cBDP (R) exhibited almost mirror-image CD signals due to similar coupling interactions (Fig. 1c). Ag complexes with cBDP (cBDP (S) were used in the following) can distinguish the CD wavelength positions of surface plasmon resonances and cBDP excitations (Fig. 1d). The BODIPY core and (S)-BINOL moiety generated characteristic CD signals at ≈504 and 340 nm[44], respectively. Thus, the weak CD signals of Ag complexes near 400 nm were attributed to induced chiral Ag plasmons, while the positive CD band near 520 nm originated from cBDP ensemble excitations. Important spectral features in common between the CD spectra of Ag and Au complexes can be clearly observed (Fig. 1d, e). First, the positive CD bands of both complexes were redshifted compared to that of cBDP, and those of the 7nt and 2nt complexes were more redshifted than those of their 14nt counterparts. In addition, the CD intensity of both complexes increased with decreasing inter-distance between the cBDP and the nanoparticle surface. In particular, symmetry (2nt and 14nt complexes) and dissymmetry (7nt complexes) of the spectral lineshape of the positive CD band were observed. These observations suggested tunable superpositions of multiple chirality origins via site-specific control of cBDP on nanoparticle surfaces.

### Frenkel-CT mixed intermolecular coupling on confined surface

The surface concentration (denoted as $\rho$ in Fig. 2) and separation of cBDP molecules on AuNPs were thoroughly tuned to analyze intermolecular couplings (Fig. 2a–f). The positive CD band of the 7nt complexes reproducibly appeared asymmetric or even became split (Fig. 2g). This split can be attributed to excitonic correlations between cBDP molecules at close proximity on the AuNP surface[47]. The asymmetric split may arise from the overlap of the intrinsic positive CD band of cBDP with the bisignate exciton-coupled CD band. These two origins of chirality may also lead to asymmetry of the CD lineshape rather than obvious splitting with a change in the surface separation and concentration of cBDP. Comparatively, the CD lineshape of the 2nt complexes, which presumably had a higher excitonic coupling strength due to closer proximity, remained symmetric (Fig. 2h), indicating more complex CD origins. The CD lineshape of the 14nt complexes was also symmetric, mainly due to the intrinsic chirality of cBDP at large intermolecular distances (Supplementary Figs. 23a and 31). Thus, distance-dependent intermolecular couplings can be considered to cause these differences in the shapes of the CD lines.

Furthermore, we quantitatively evaluated the influence of cBDP surface concentration and separation on CD enhancement (Supplementary Figs. 22 and 23). As shown in Fig. 2i, with increasing cBDP surface concentration, significant increases in the anisotropic dissymmetry factors (g-factors) were observed for all the complexes. In addition, we measured the CD enhancement factor of the BODIPY core (CD$_{complex}$/CD$_{cBDP}$) near 520 nm and the internal ratio of

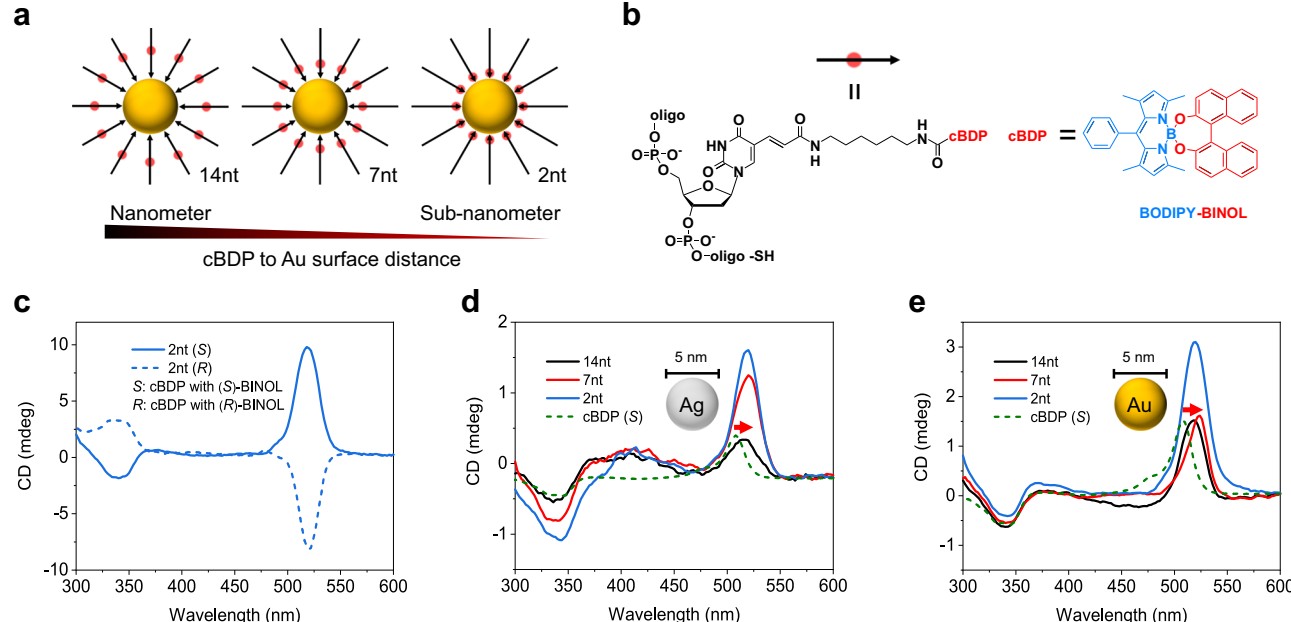

**Fig. 1 | Structures and optical properties of site-specific hybrid complexes.**
**a** Schematic of site-specific hybrid complexes. The cBDP is represented by a red spot and the DNA backbone is represented by a black arrow. **b** Structural formula of cBDP-DNA conjugates. The electron-withdrawing BODIPY core is highlighted in blue, and the electron-donating BINOL moiety is highlighted in red. The designation "II" stands for "equal to". **c** CD spectra of Au complexes with cBDP (*S*) and cBDP (*R*). **d**, **e** CD spectra of cBDP (*S*), Ag (**d**), and Au (**e**) complexes with cBDP (*S*). The cBDP surface concentrations in the three complexes were close in (**d**) and (**e**), respectively. The red arrow indicated a clear redshift. Source data are provided as a Source Data file.

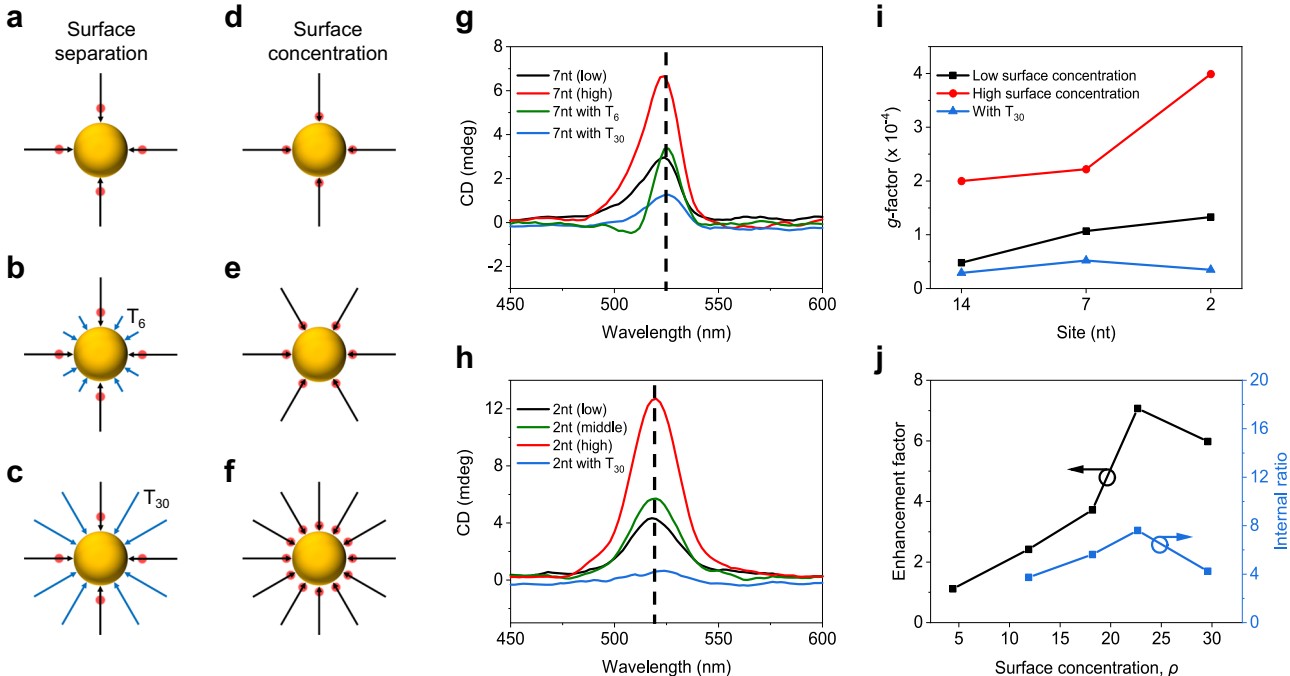

**Fig. 2 | Influence of intermolecular interactions within cBDP ensembles. a–f**
Schematic of tuning of surface separation and concentration of cBDP. The cBDP is represented by a red spot, and the DNA backbone is represented by a black arrow. $T_6$ and $T_{30}$ stand for DNA spacer co-modified on AuNPs and are represented by blue arrows. **g–i** CD spectra of 7nt (**g**) and 2nt (**h**) complexes, and *g*-factors (**i**) of 14nt, 7nt, and 2nt complexes with varying surface separations and concentrations of cBDP. **j** CD enhancement factor and internal ratio of 2nt complexes with varying surface concentrations of cBDP. "Low" refers to cBDP/AuNP molar ratio during synthesis, i.e., $n_{cBDP}:n_{Au} = 25:1$, "middle" refers to $n_{cBDP}:n_{Au} = 50:1$, and "high" refers to $n_{cBDP}:n_{Au} = 100:1$ (see Supplementary Tables 1–3 for details). Co-modification of AuNPs with $T_{30}$ resulted in the lowest molar ratio. Surface concentration, $\rho$, is the number of cBDP-DNA per AuNP without a unit. Source data are provided as a Source Data file.

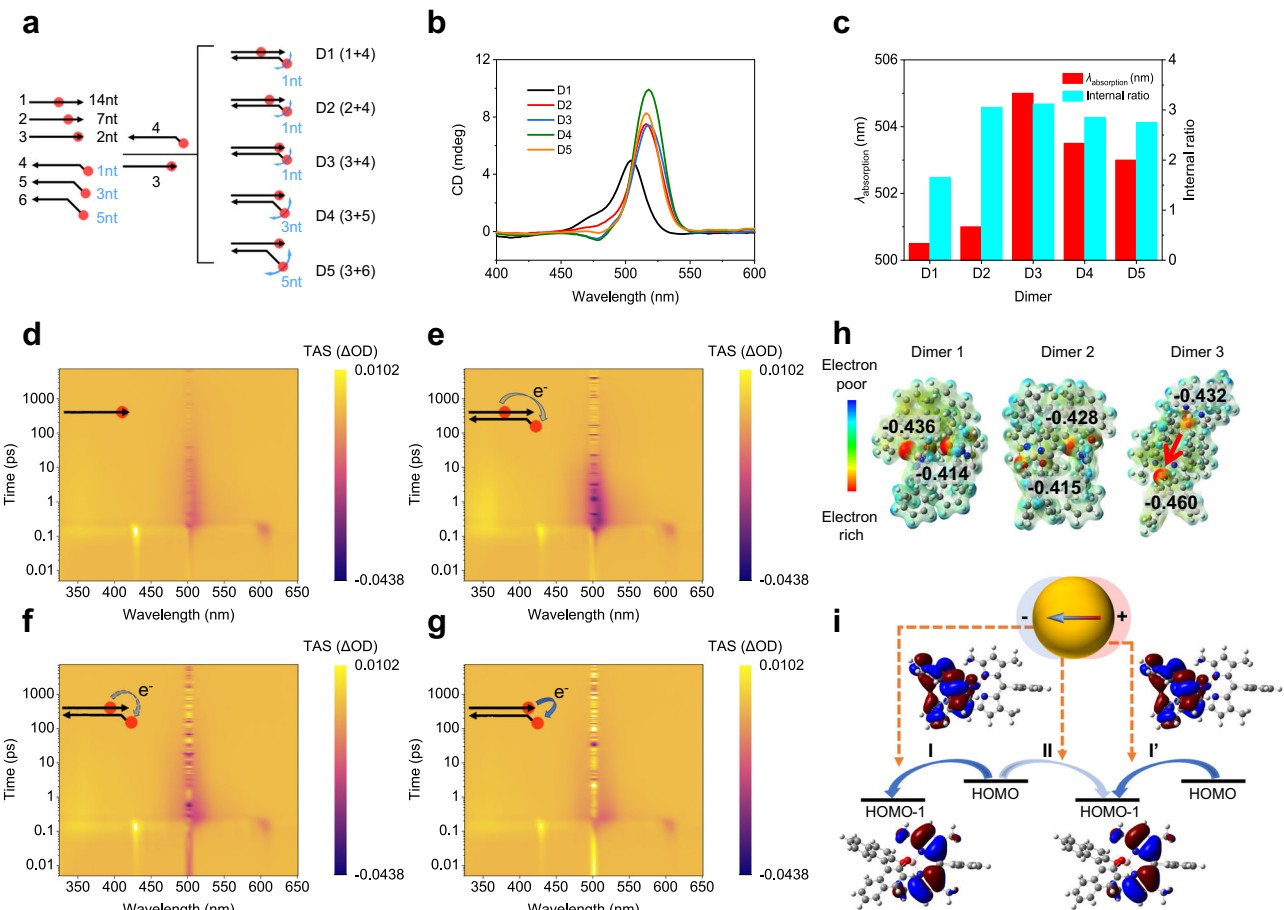

**Fig. 3 | Construction and intermolecular interactions of cBDP dimers.**
**a** Schematic of DNA templating of cBDP dimers. The cBDP is represented by a red spot and the DNA backbone is represented by a black arrow. The blue curved arrows depict the explored space of the corresponding cBDP molecule. **b, c** CD spectral evolution (**b**), maximum absorption wavelength, and CD internal ratio of cBDP dimers (**c**). **d–g** Color image plots of TA experiments of cBDP-DNA monomer (**d**) and D1–D3 (**e–g**) at the same timescale. The purple color corresponds to the ground state bleaching (GSB) band at ≈504 nm. **h** DFT calculated charge density distribution map of the cBDP dimer with three different spatial orientations (Supplementary Fig. 39, Supplementary Data 2). The red arrow in Dimer 3 shows apparent intermolecular CT. **i** Interference of surface localized plasmonic field with intra- and intermolecular CT via altering electronic couplings during electron migration. The red (blue) color indicates the isosurfaces of the wave function with a positive (negative) sign. Source data are provided as a Source Data file.

CD intensity between the BODIPY core and the (*S*)-BINOL moiety ($|CD_{520\,nm}|/|CD_{340\,nm}|$) (Fig. 2j and Supplementary Fig. 25). The CD enhancement factor of the 2nt complexes first increased rapidly with increasing cBDP surface concentration and then decreased after reaching a maximum (Fig. 2j and Supplementary Fig. 25). Similarly, the change in the CD internal ratio followed this trend. Comparatively, the internal ratios for the 7nt (3.2) and 14nt (2.4) complexes were greater than that of the cBDP molecule (≈1.8, Fig. 1d, e and Supplementary Fig. 14) but much lower than that of the 2nt complex (7.6) at the approximate cBDP surface concentration (Supplementary Fig. 23 and Supplementary Table 3). These observations strongly suggested the critical role of distance-dependent intermolecular couplings in CD enhancement.

As shown in Fig. 2j, at the lowest cBDP surface concentration where the intermolecular couplings are negligible, CD enhancement was hardly observed, demonstrating that there was no obvious interfacial CT between the AuNPs and the cBDP molecules or that the interfacial CT had little effect on the CD intensity. Increasing the surface concentration of cBDP can reduce intermolecular distances, thus enhancing Coulombic interactions and even provoking molecular contact that generates orbital overlap and intermolecular CT. The latter was reported to significantly affect the chiroptical responses of molecular ensembles[48,49]. These multiple intermolecular interactions peaked at the threshold of the cBDP surface concentration, beyond which cBDP overcrowding could result in unfavorable molecular stacking that hampers CD enhancement[24,50–54]. Noticeably, the decrease in the CD enhancement factor after the threshold suggested that the CD signals at ≈520 nm mainly originated from cBDP ensemble excitations and interactions rather than from induced chiral plasmons because the induced CD intensity would be positively correlated with the amount of cBDP[10,24,38,55–57].

To further analyze the intermolecular couplings, we investigated the optical responses of pure molecular aggregates of cBDP without a plasmonic background. Discrete cBDP dimers were assembled as models via a DNA sequence-selective templating strategy (Fig. 3a and Supplementary Fig. 27). This method allows programmable design and accurate control of intermolecular distances and conformational dimensions of the cBDP dimers. As depicted in Fig. 3a, '14nt', '7nt', and '2nt' site-specific control of cBDP in strands 1–3 determined the intermolecular distance in D1–D3. Meanwhile, '1nt', '3nt', and '5nt' unpaired bases in strands 4–6 dictated the conformational freedom of D3–D5. At large intermolecular distances (D1), the CD lineshape and internal ratio resembled those of pure cBDP molecules (Fig. 1c, d and Supplementary Fig. 14), demonstrating little intermolecular coupling (Fig. 3b, c). As the distance decreased, e.g., from dimers D1 to D3, the original positive CD band evolved into an asymmetric splitting lineshape (Fig. 3b). Similarly, D3–D5 also showed distance-dependent CD

splitting. Such CD splitting, a phenomenon of Davydov splitting[58,59], similar to that in 7nt complexes (Fig. 2g), arises from coherent excitonic correlations between cBDP molecules. The CD internal ratio variation and absorption peak shift well reflected the changes in intermolecular distance and conformational dimension, which are closely related to Davydov splitting (Fig. 3c).

Femtosecond transient absorption (TA) spectroscopy was employed to examine the intermolecular CT in the cBDP dimers, as shown in Fig. 3d–g. In the color image, the purple color corresponds to the ground state bleaching (GSB) band at ≈504 nm. cBDP exhibited a negative GSB band that is associated with the $S_0 \rightarrow S_1$ transition[60]. The GSB decay of D1 (Fig. 3e) was quite similar to that of the monomer, consistent with the above CD analysis. We performed the kinetic analysis with exponential fittings of the decay curves at 514 nm to avoid interference from laser incidence at 504 nm (Supplementary Fig. 28). Dimers D2 and D3 proved intermediate states with a timescale of ≈4 ps (Supplementary Table 4), in contrast to D1 and the monomer. This difference indicated the existence of intermediate excited states, likely caused by intermolecular CT in closely spaced cBDP dimers[60–64].

Density functional theory (DFT) calculations were also performed on the cBDP dimers (Supplementary Fig. 39, Supplementary Data 2). When two cBDP molecules were spatially close to each other with appropriate relative orientation, intermolecular CT occurred and resulted in an apparent change in the charge density distribution (Fig. 3h, Supplementary Data 2). DFT calculations also showed that CT from the highest occupied molecular orbital (HOMO) (mainly located in the (S)-BINOL moiety) to the low-lying semivacant HOMO-1 (mainly located in the BODIPY core) was thermodynamically favorable within the molecule itself (Process I and I') or between neighboring molecules (Process II) (Fig. 3i)[60,61]. Multiple CT among cBDP ensembles, not limited to cBDP dimers, might occur when cBDP is densely stacked on the confined surfaces of AuNPs. This multiple CT may be affected by the localized plasmonic field on the AuNP surface (Fig. 3i).

## Plasmon-altered collective chiroptical response

The influence of the localized plasmonic field was systematically studied by varying the size of AuNPs with cBDP site-specificity and assembling superstructures with controllable positioning of cBDP. As illustrated in Fig. 4a, b, when the cBDP-AuNP inter-distance decreased from 14nt to 2nt, the g-factors of the complexes with AuNP diameters varying from 3 to 40 nm increased consistently. Noticeably, as the AuNP size decreased, the $g_{2nt}/g_{14nt}$ ratio also increased. Theoretical analysis based on the dipole-coupling approximation successfully captured the increasing trend of the g-factor ratio $g_{2nt}/g_{14nt}$ (Fig. 4c). The spatial distribution of the surface electric field on single AuNP was also calculated, revealing an increase in intensity as the cBDP–AuNP inter-distance decreased (Fig. 4d and Supplementary Fig. 41). The smaller the particle size is, the faster the increase in the electric field intensity near the surface. Evidently, the changing trend of the $g_{2nt}/g_{14nt}$ ratios follows the spatial variation in the electric field distribution of differently sized AuNPs.

The discrepancy between the observed increasing rate of g-factors and the theoretical prediction (Fig. 4b, c), especially for 3 nm, 7nt complexes, was mainly attributed to the complex intermolecular couplings in the cBDP ensembles, as discussed above. The spectral linewidth of the dominant CD signals near 520 nm of the single-particle complexes is similar to that of the cBDP molecules at ≈504 nm (Supplementary Figs. 14 and 31). This again demonstrated the minor contribution of induced plasmonic CD, which would have broadened the spectral linewidth due to the lossy nature of plasmonic resonances and hence the major contribution of molecular CD, which was affected by Frenkel-CT mixed intermolecular couplings. Taken together, the complex intermolecular couplings of cBDP ensembles could be correlated with the spatial inhomogeneity of the local electric field distribution of AuNPs under the dipole approximation limit.

Figure 4e–l shows the experimental and theoretical results of the superstructures. Figure 4e depicts the structural model of the superstructures that were built via DNA-hybridization between the complementary core and satellite particles (see Supplementary Tables 6 and 7 for DNA sequences). Figure 4f shows the small-angle X-ray scattering (SAXS) profiles of the superstructures, which are similar to previous results[42]. The stability and mono-dispersity of these superstructures were verified by gel electrophoresis and dynamic light scattering (DLS) (Supplementary Figs. 35 and 38). The extinction peak wavelength of these superstructures redshifted by ≈12 nm relative to that of the constituent core particles, and the wavelength position of the CD peak correspondingly shifted to the red (Fig. 4g, h). This agreement of CD spectral shift suggested the emergence of induced plasmonic chirality as a result of the backward interactions from the cBDP ensembles. The CD peak intensity was more than 6.7 times greater than that of solely satellite complexes. The control superstructures without cBDP displayed no observable CD signals, thus ruling out possible structural chirality (Fig. 4h and Supplementary Fig. 37). Because the cBDP molecules were largely separated from the core particle surface (30 base pairs of DNA linkers, ≈10 nm), the induced plasmonic CD of the core particles was negligible. Therefore, the observed CD signals mainly originated from the satellite complexes but were enhanced by the presence of core particles.

Electromagnetic simulations revealed a U-shaped curve of the electric field intensity in the core-satellite gap, with a large field enhancement near the satellite particles (Fig. 4i, j). Furthermore, we compared the field enhancement of satellite particles in the superstructure (black line in Fig. 4k) with that in the pure satellite complexes (blue line in Fig. 4k) through the integration and averaging of the local electric field of a spherical surface surrounding the satellites (2nt surface distance, Supplementary Fig. 42). Apparently, the average field intensity of the satellites in the superstructure is much greater than that without coupling to the core (Fig. 4k and Supplementary Figs. 42–45). The field intensity in the 2nt superstructure is much stronger than that in the 7nt and 14nt counterparts (Fig. 4l). Therefore, such electric field enhancement resulting from core–satellite plasmonic couplings could account for the CD enhancement of the superstructure. These findings consistently corroborated the strong association of the collective chiral responses of the superstructure with the spatial inhomogeneity of the local electric field of the AuNPs. Also, the Frenkel–CT mixed intermolecular couplings of cBDP could be greatly modified by localized surface plasmons. Additionally, DFT calculations revealed that the external electric field indeed significantly altered the charge density distribution of cBDP (Supplementary Fig. 40), which could further affect the intermolecular CT via electronic couplings[65].

## Discussion

In summary, we experimentally generated chiral plasmonic hybrid complexes with precise control of molecular site-specificity at the nanometer to sub-nanometer scale and explored the role of energy and electron transfer in chiral generation and amplification. Joint experimental and theoretical studies revealed the essential role of intermolecular interactions between Frenkel–CT mixed excited states. This Frenkel–CT mixing appeared to be modified by the localized surface plasmons of the nanoparticles. The results represent a departure from the idea of describing chiral molecular excitations in localized plasmonic fields without taking coherent excitonic correlations and CT processes into account. Thus, this study provides experimental models and evidence for advancing the current theoretical framework, especially considering supramolecular chiral plasmonic systems under

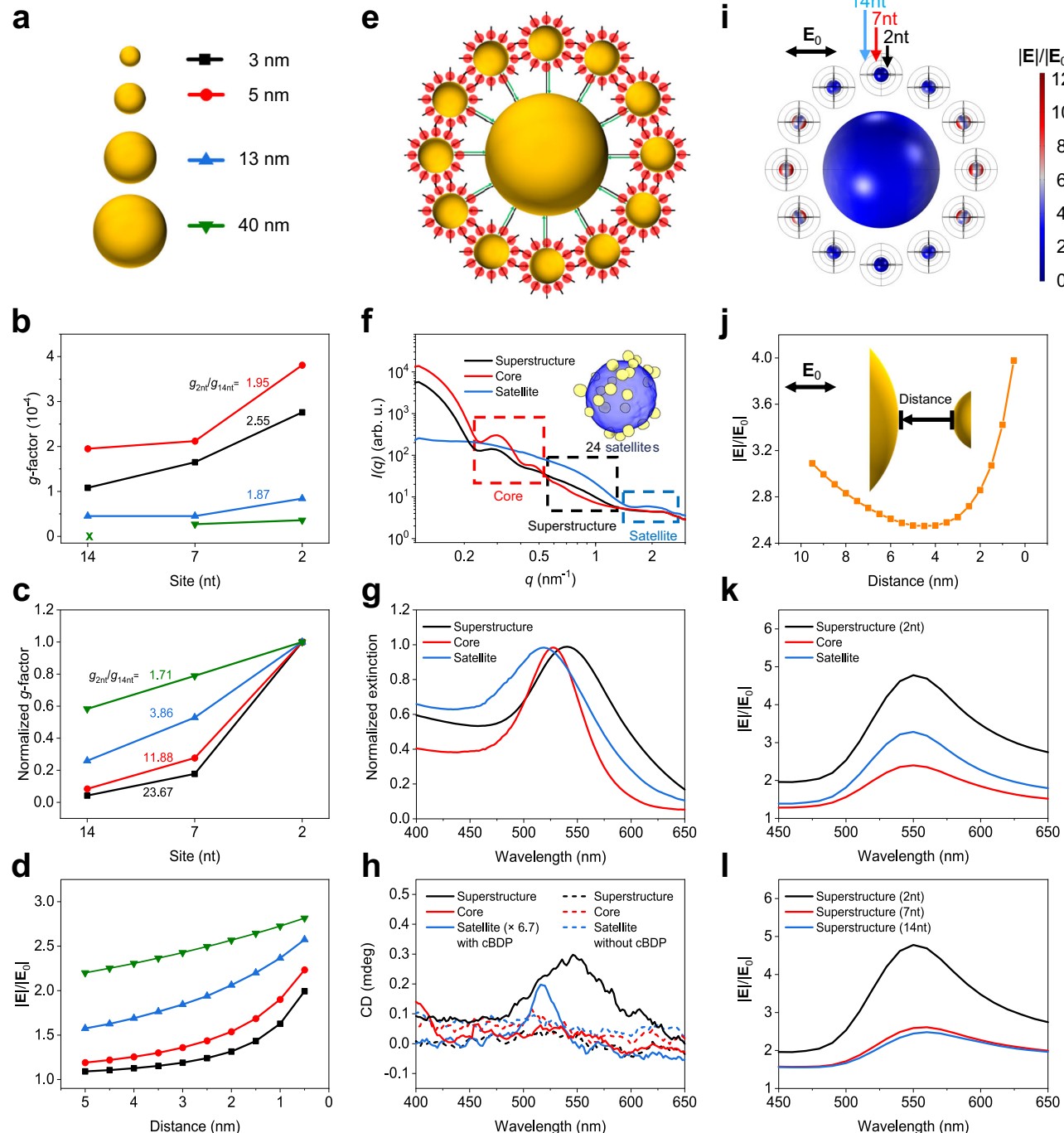

**Fig. 4 | Influence of localized surface plasmons of nanoparticles. a–d** Schematic of size variation (**a**), experimental (**b**), and theoretical (**c**) *g*-factors of single-particle complexes, and electric field intensity as a function of distance from the surface of differently sized AuNPs (**d**). **e**, **f** Schematic (**e**) and small angle X-ray scattering (SAXS) profiles (**f**) of superstructures. The cBDP is represented by a red spot, and the DNA linkers are represented by black and green arrows. The inset shows a three-dimensional (3D) tomographic reconstruction of a single superstructure (Supplementary Data 3, Supplementary Movie 1). **g**, **h** Extinction and CD spectra of the superstructures and the constituents, respectively. The designation "Satellite" means all small particles (an estimated number of 24) surrounding the core particle. **i**, **j** Electric field intensity on the particle surface (**i**) and in the core-satellite gap (**j**), excited in the horizontal direction. **k** Comparison of average $|E|/|E_0|$ of satellite particles in superstructures (2nt) and pure satellite complexes (2nt). Core complexes (14nt) were used as a control. **l** Average $|E|/|E_0|$ of satellite particles in different superstructures. Source data are provided as a Source Data file.

the collective coupling regime and treating multiple degrees of freedom (photonic, electronic, and nuclear/ionic). Moreover, the collective interactions of Frenkel−CT excited states and plasmon resonances provide opportunities for harnessing spin-charge transport for various applications, for example, in asymmetric catalysis and photon spin discrimination.

## Methods

### Synthesis and characterization of AuNPs

3 nm AuNPs were synthesized by the reduction of tetrachloroauric acid (HAuCl$_4$) with tannic acid (C$_{76}$H$_{52}$O$_{46}$) and sodium citrate (C$_6$H$_5$Na$_3$O$_7$) following published procedures[66]: To a mixed aqueous solution containing 150 mL of sodium citrate (2.2 mM) and 0.1 mL of tannic acid

(2.5 mM) at 70 °C, 1 mL of HAuCl$_4$ (25 mM) aqueous solution was added. The solution turned from transparent to dark gray instantaneously after the gold precursor addition, and then to brownish-orange within a few minutes.

5 nm AuNPs were synthesized by a sodium citrate/tannic acid method following published procedures[67]: To a mixed aqueous solution containing 11 mL of sodium citrate (50 mM), 1.7 mL of potassium carbonate (150 mM) and 16.7 μL of tannic acid (25 mM) at room temperature, 1.6 mL of HAuCl$_4$ (25 mM) aqueous solution was added. The solution turned from transparent to dark gray instantaneously after the gold precursor addition and then to red within 24 h.

13 and 40 nm AuNPs were synthesized by Fren's method and seeded growth process following published procedures[68], respectively. 13 nm AuNPs: To a boiling aqueous solution containing 50 mL of HAuCl$_4$ (1 mM), 5 mL of sodium citrate (1% w/v) aqueous solution was added under vigorous stirring. Stirring was stopped after 15 min, and the solution was cooled slowly to room temperature, resulting in the preparation of 13 nm AuNPs. 40 nm AuNPs: To an aqueous solution containing 50 mL of 13 nm AuNPs (2 nM), 980 μL of sodium citrate (1% w/v), 980 μL of HAuCl$_4$ (24.2 mM), and 980 μL of ascorbic acid (1% w/v) are added separately over 1 h. The mixture was heated to boiling and maintained at that temperature for 30 min. The solution was cooled down slowly to room temperature, resulting in the preparation of 20 nm AuNPs. Then, to an aqueous solution containing 50 mL of 20 nm AuNPs (2 nM), 9.5 mL of sodium citrate (1% w/v), 9.5 mL of HAuCl$_4$ solution (24.2 mM), and 9.5 mL of ascorbic acid (1% w/v) are added separately over 1 h. The mixture was heated to boiling and kept for 30 min. The solution was then cooled down slowly to room temperature, resulting in the preparation of 40 nm AuNPs.

### Synthesis and characterization of BODIPY-BINOL NHS ester

The synthesis of BODIPY NHS ester is according to reference with slight modifications[69]. A typical procedure is as follows: To a solution of BODIPY-COOMe (148.0 mg, 0.387 mmol) in isopropanol (15 mL) under a nitrogen atmosphere, an aqueous solution of 0.1 m KOH (15 mL) was added. The reaction mixture was stirred at room temperature until complete consumption of BODIPY-COOMe by thin layer chromatography (TLC) analysis was observed. The reaction mixture was then acidified with dilute HCl until pH = 2–3, filtered and evaporated to dryness *in vacuo*. The residue was purified by silica gel gradient column chromatography (CH$_2$Cl$_2$/ethyl acetate, 5/1 to 1/5, v/v) to afford the BODIPY-COOH (60.4 mg). The crude product was used directly for esterification.

To a solution of BODIPY–COOH (60.4 mg) in CH$_2$Cl$_2$ (15 mL) was added *N*-hydroxy-succinimide (NHS) (37.8 mg, 0.328 mmol), 1-(3-dimethylaminopropyl)-3-ethylcarbodiimide hydrochloride (EDCI) (62.9 mg, 0.328 mmol), and the solution was stirred in the dark overnight. After completion of the reaction by TLC analysis, the mixture was diluted with water, extracted with CH$_2$Cl$_2$ (2 × 25 mL), and washed with brine (1 × 25 mL). The combined organic phase was dried over anhydrous MgSO$_4$, filtered and concentrated in vacuo. Purification of the crude residue by silica gel gradient column chromatography (hexane to CH$_2$Cl$_2$) afforded BODIPY NHS ester (48.0 mg, 0.103 mmol) as orange-red solid. $R_f$ = 0.80 (MeOH/ethyl acetate, 1/4, v/v); $^1$H NMR (400 MHz, Chloroform-*d*) δ 8.28 (d, *J* = 8.4 Hz, 2H), 7.51 (d, *J* = 8.3 Hz, 2H), 6.01 (s, 2H), 2.96 (s, 4H), 2.57 (s, 6H), 1.38 (s, 6H).

The synthesis of BODIPY-(*S*)-BINOL NHS ester is as follows: To a solution of BODIPY NHS ester (48.0 mg, 0.103 mmol) and 1,1'-binaphthyl-2,2'-diol ((*S*)-BINOL) (147.7 mg, 0.515 mmol) in dry CH$_2$Cl$_2$ (15 mL) was added dropwise Et$_2$AlCl (1.00 M in hexane, 206 μL, 0.206 mmol) under nitrogen atmosphere and the solution was stirred in the dark for 2 h. After the solvent was removed by rotary evaporator, the crude residue purified by silica-gel column chromatography (CH$_2$Cl$_2$/hexane, 3/1, v/v) afforded BODIPY-(*S*)-BINOL NHS ester (42.3 mg, 0.059 mmol) as red solid[44,60,70]. $R_f$ = 0.68 (CH$_2$Cl$_2$); $^1$H NMR (400 MHz, chloroform-*d*)

δ 8.30 (d, *J* = 8.3 Hz, 2H), 7.84 (d, *J* = 8.3 Hz, 2H), 7.79 (d, *J* = 8.7 Hz, 2H), 7.60 (d, *J* = 8.3 Hz, 2H), 7.32 (ddd, *J* = 8.1, 6.7, 1.3 Hz, 2H), 7.24 (d, *J* = 8.3 Hz, 2H), 7.21 (d, *J* = 8.7 Hz, 2H), 7.15 (ddd, *J* = 8.3, 6.6, 1.4 Hz, 2H), 5.82 (s, 2H), 2.96 (s, 4H), and 1.66 (s, 6H), 1.37 (s, 6H).

The synthesis of BODIPY-(*R*)-BINOL NHS ester adopted the same procedure of BODIPY-(*S*)-BINOL NHS ester except for using 1,1'-binaphthyl-2,2'-diol ((*R*)-BINOL). $^1$H NMR (400 MHz, chloroform-*d*) δ 8.31 (d, *J* = 8.3 Hz, 2H), 7.85 (d, *J* = 8.3 Hz, 2H), 7.80 (d, *J* = 8.7 Hz, 2H), 7.61 (d, *J* = 8.3 Hz, 2H), 7.32 (ddd, *J* = 8.1, 6.7, 1.3 Hz, 2H), 7.24 (d, *J* = 8.3 Hz, 2H), 7.22 (d, *J* = 8.7 Hz, 2H), 7.15 (ddd, *J* = 8.3, 6.6, 1.4 Hz, 2H), 5.82 (s, 2H), 2.96 (s, 4H), 1.66 (s, 6H), and 1.37 (s, 6H).

### Conjugation of DNA strand and chiral chromophore

The chromophore–DNA conjugates were generally prepared by using an amine–NHS ester reaction. In a typical amine–NHS ester conjugation reaction, the amine-modified DNA was mixed with BODIPY-BINOL NHS ester and appropriate buffer. The mixture was placed on a shaker for several hours at room temperature and then purified by a reverse phase HPLC system using a C18 column (solvent A: trimethylamine acetate (TEAA) buffer, 100 mM, pH = 7.0; solvent B: acetonitrile). The collected fraction was used directly for AuNP surface modification. The resultant conjugates of chiral BODIPY (cBDP) and DNA (denoted as cBDP-DNA, i.e., cBDP (*S*)-DNA (14nt), cBDP (*S*)-DNA (7nt), cBDP (*S*)-DNA (2nt) and cBDP (*R*)-DNA (2nt)) were characterized by mass spectrometry.

### Stability test of chromophore–DNA conjugates

*pH stability test*: Two equal amounts of cBDP-DNA (2nt) (30 μL, 20 μM) were incubated for 30 min at room temperature in 400 μL sodium citrate–HCl buffer (500 mM sodium citrate, pH = 3.0) and 400 μL 0.5 × TE buffer (5 mM Tris, 0.5 mM EDTA, pH = 8.0), respectively.

*Tris(2-carboxyethyl)phosphine hydrochloride (TCEP·HCl) stability test*: Two equal amounts of cBDP-DNA (2nt) (40 μL, 23 μM) were incubated for 30 min at room temperature in 400 μL 0.5 × TE buffer (pH = 8.0) and 396 μL 0.5 × TE buffer (pH = 8.0) supplemented with 4 μL TCEP·HCl solution (200 mM), respectively.

*Dithiothreitol (DTT) stability test*: Two equal amounts of cBDP-DNA (2nt) (16 μL, 25 μM) were incubated overnight at room temperature in 500 μL 0.5 × TE buffer (pH = 8.0) and 498 μL 0.5 × TE buffer (pH = 8.0) supplemented with 2 μL DTT solution (1 M), respectively.

### Synthesis of single-particle hybrid complexes

AuNPs were modified with thiolated DNA strands following published protocols with slight changes[71]. Typically, 800 μL of ≈25 nM citrate-capped 5 nm AuNPs solution was mixed with 80 μL of 25 μM cBDP-DNA (2nt) pretreated with 2.5 μL of 200 mM TCEP. The above mixture was frozen at −20 °C overnight and then thawed at room temperature. After that, the complexes were concentrated by centrifugation. Finally, the resulting products were purified by gel electrophoresis in 0.5 × TBE buffer (44.5 mM Tris, 44.5 mM boric acid, 1 mM EDTA, pH = 8.0) for further characterization.

### Quantification of surface concentration of chromophore–DNA conjugates

The hybrid complexes were incubated with DTT (12.5 mM for all samples) at 30 °C overnight to displace cBDP-DNA from the AuNP surface. The concentrations of AuNPs and cBDP-DNA were obtained by their absorption, i.e., $A = \varepsilon_1 b c_1$, and $CD = \Delta\varepsilon_2 b c_2$, where $\varepsilon_1$ is the molar extinction coefficient of AuNPs, $\Delta\varepsilon_2$ is a difference of molar extinction coefficient of cBDP-DNA, $\Delta\varepsilon_2 = \varepsilon_{LCP} - \varepsilon_{RCP}$, $b$ is the light pass length, $c_1$ and $c_2$ are concentrations of AuNPs and cBDP-DNA, respectively (Supplementary Fig. 22). As exemplified in Supplementary Fig. 23, the difference in absorption spectra near 520 nm before and after the addition of DTT can be used to estimate the concentration of AuNPs ($c_1$), and the CD intensity at 504 nm after the addition of DTT can be

used to estimate the concentration of displaced cBDP-DNA ($c_2$). The number of cBDP-DNA per AuNP, i.e., surface concentration, is obtained by $\rho = c_2/c_1$, without unit.

## Self-assembly of cBDP molecular dimers

Two complementary cBDP-DNA conjugates were added together in equal molar amounts into $0.5 \times$ TE buffer (pH = 8.0) with 300 mM NaCl. The sample was annealed by cooling from 60 to 30 °C (1 °C per 20 min) with a final concentration of 2.5 μM, then used directly for further characterization.

## Self-assembly of superstructures

The core particles (40 nm AuNPs co-modified with $A_{30}$ and spacer $(ACT)_6$) were combined with an excessive amount of satellite complexes (5 nm, 2nt) in $0.5 \times$ TAE·Mg buffer (20 mM Tris, 20 mM acetic acid, 1 mM EDTA, and 6.25 mM magnesium acetate, pH = 8.0) to form the superstructures. The DNA-hybridization-induced self-assembly of superstructures took place by an annealing procedure from 60 to 30 °C (1 °C per 20 min). 1 wt% agarose gel electrophoresis was employed to isolate the as-formed superstructures.

## Circular dichroism and absorption measurements

The CD spectra were measured using a Chirascan V100 CD spectrometer, scanning from 300 to 700 nm with an incident bandwidth of 1 nm, a path length of 1 cm, and a scan speed of 200 nm min$^{-1}$. Absorption spectra were collected on the same spectrometer for simultaneous correlations with the CD spectra. $0.5 \times$ TBE buffer (for single-particle complexes), 300 mM NaCl–$0.5 \times$ TE buffer (for cBDP dimers), and $0.5 \times$ TAE·Mg buffer (for superstructures) were measured as the background for the baseline correction of the CD and absorption spectra.

## Femtosecond and nanosecond transient absorption spectroscopy

Titanium sapphire femtosecond laser (Astrella, Coherent Inc.) produced femtosecond pulse light with a center wavelength of 800 nm, a repetition rate of 1 kHz, and a pulse width of 100 fs. The 800 nm pulse light was divided into two parts by the beam splitter. One part entered the optical parametric amplifier (OPerA Solo, Coherent Inc.) to generate the pump light of the required wavelength for the experiment, and the other part passed through the delay line and entered the transient absorption spectrometer (Helios, Ultrafast system), and focused on the nonlinear calcium fluoride crystal to generate a supercontinuous probe light from 320 to 640 nm. Polarization between the pump and the probe was set to the magic angle (54.7°). The instrumental response is ≈120 fs. Samples were dissolved in 300 mM NaCl–$0.5 \times$ TE buffer (pH = 8.0). The steady-state absorbance at the excitation wavelength (504 nm) was ≈0.1 OD in a 2 mm quartz cuvette. The CD and steady-state absorption spectra were monitored before and after the TA experiments to examine if there is non-negligible photodegradation.

## Small-angle X-ray scattering

SAXS measurements were performed on beamline BL16B1 of the Shanghai Synchrotron Radiation Facility. The instrument was equipped with an X-ray source with a wavelength of 1.24 Å and an area detector consisting of $3 \times 8$ panels (Pilatus 2M). 2-D SAXS data were calibrated using a silver behenate standard. Saxsgui v2.15.01 was used to generate the 1-D integrated scattering profiles of intensity $I$ versus scattering vector $q$.

## Tomography data acquisition

Electron tomography tilt series was performed at 200 kV using a Talos Arctica (Thermo Fisher Scientific) equipped with a K2 Summit direct detector camera (Gatan). Tilt series acquisition was driven by SerialEM software[72] using dose-symmetric tilt-scheme[73] with a tilt increment of 2°, spanning an angle range from +52° to −52° (Supplementary Data 3, Supplementary Movie 1). The TEM magnification corresponding to a camera pixel size of 0.92 Å was used, and the target defocus was set to −1 μm. The total dose for a tilt series was ≈3794 e Å$^{-2}$.

## Tomogram reconstruction and segmentation

The Motioncor2[74] motion-corrected tilt images were aligned with the patch-tracking method and reconstructed to obtain the tomograms by the back-projection algorithm in the IMOD 4.11.0 software package[75]. The segmentation was built by IMOD 4.11.0 software package[75] and rendered by Chimera X[76].

## Quantum chemical calculation

All the calculations were performed with Gaussian16 in the framework of the density functional theory with the B3LYP method. The 6-311G(d,p) basis set is used for the geometry optimization (C, H, O, N, and B atoms). The long-range van der Waals interaction is described by the DFT-D3 approach[77]. The optimal structure was calculated using the PCM solvent model in an aqueous solution. Molecular structures were visualized using GaussView 6.1.1. For a given geometry, the charge distribution in the ground state was derived from the Mulliken charges on the donor (($S$)-BINOL moiety) and the acceptor (BODIPY core). In addition, we applied an electric field of magnitude 0.1 V Å$^{-1}$ in the direction parallel or perpendicular to the molecular principal axis to analyze the electron transfer within the molecule.

## Electromagnetic simulation

Our generic model incorporated a 40 nm AuNP encircled with 12 AuNPs having 5 nm diameter in a superstructure, built into a homogeneous matrix with an effective dielectric constant, $\varepsilon_{\mathrm{eff}} = 2$. The satellites and core particles lay on the same plane ($xy$ plane). The center-to-center distance between the 40 nm core particle and 5 nm satellites was 32.5 nm, accordingly, the core-satellite surface distance was 10 nm. We used COMSOL Multiphysics® to compute the electromagnetic response of three objects: core (40 nm AuNP), satellite (5 nm AuNP), and a superstructure. We used a linearly polarized electromagnetic wave with polarization along the $x$-axis. The impinging wave was parallel to the $z$-axis, which was orthogonal to the plane of the superstructure. Particularly, we computed the electric field enhancement averaged on a spherical surface with different distances from the AuNP surface (Fig. 4i). These distances corresponded to the length of DNA linkers (2nt, 7nt, 14nt).

For unique structures, such as a 5 nm and a 40 nm AuNP, which are the building elements of the superstructure, the formalism of the averaged electromagnetic enhancement is given as follows:

$$|\mathbf{E}|_{\mathrm{av}} = \frac{1}{S_{\mathrm{NP,i}}} \int_{S_{\mathrm{NP,i}}} \frac{\sqrt{\mathbf{E} \cdot \mathbf{E}^*}}{E_0} \tag{1}$$

$\mathrm{NP}_i$ is either a 40 nm or a 5 nm AuNP, $\mathbf{E}$ and $\mathbf{E}^*$ are electric fields and its complex conjugate on the integrated surface ($S_{\mathrm{NP},i}$). $S_{\mathrm{NP,i}}$ is the spherical surface near the AuNP on which the norm of the electric field is integrated. $E_0$ is the amplitude of the incident electromagnetic field.

The enhancement of the electric field for the superstructure is given by the following:

$$|\mathbf{E}|_{\mathrm{av}} = \frac{1}{N_{\mathrm{tot}} S_{\mathrm{NPs}}} \sum_{\substack{\text{all} \\ \text{satellite} \\ \text{NPs}}} \int_{S_n} \frac{\sqrt{\mathbf{E} \cdot \mathbf{E}^*}}{E_0} \tag{2}$$

$N_{\mathrm{tot}}$ is the total number of satellites and $S_{\mathrm{NPs}}$ is the spherical surface, which is used for the electric field norm integration.

To estimate *g*-factor depending on the distance between cBDP molecules and AuNP, we used the following analytic form originating from dipole–dipole interactions.

$$g_{\mathrm{nt}} = \mathrm{const}\, \frac{N_{\mathrm{dye}} r^3}{(r + d)^3} \qquad (3)$$

where $N_{\mathrm{dye}}$ is the number of dye molecules, $r$ is the radius of AuNP, and $d$ is the distance of the molecule from the AuNP surface. Since we cannot exactly determine the number of molecules, we used the approach of the normalized (to 1) *g*-factor (Fig. 4c). In this manner, one may see the evolution of the *g*-factor depending on the AuNP diameter and distance of chiral molecules from them.

### Reporting summary

Further information on research design is available in the Nature Portfolio Reporting Summary linked to this article.

## Data availability

The data that support the findings of this study are available from the corresponding authors upon request. The NMR data are available from Figshare[78]. Source data are provided with this paper.

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

## Acknowledgements

This research is supported by the National Key Research and Development Program of China (2021YFA1200300 X.L.), the National Natural Science Foundation of China (22372031 X.L.), the Natural Science

Foundation of Shanghai (21ZR1400700 X.L.) and the Spanish MICINN (Grant PID2020-114755GB-C32 S.d.l.M.). We thank Prof. Xunda Feng for the assistance in the SAXS characterizations done on beamline BL16B1 at the Shanghai Synchrotron Radiation Facility. We acknowledge the Cryo-EM Platform of Peking University for assistance with TEM. We also thank Dr. Menghui Jia at the Materials Characterization Center, ECNU Multi-functional Platform for Innovation, for supporting transient absorption spectroscopy characterization.

## Author contributions

The project was conceived and supervised by X.L. The experiments were carried out by H.L., X.X., Z.L., Q.X., Z.J., Y.Y., M.K., Q.G. Experimental data analysis was performed by X.L., H.L., and X.X. Theoretical calculations were conceived by J.W., H.H., A.G. and performed by R.G. and A.M. The manuscript was written by X.L. and H.L. All authors, including H.W., S.d.l.M., G.M., Z.W., T.Y., and Z.T., contributed to discussions about the results.

## Competing interests

The authors declare no competing interests.
