## [Peer Review File · Nature Communications]

Collective chiroptical activity through the interplay of excitonic and charge-transfer effects in localized plasmonic fieldsReviewer #1 (Remarks to the Author):

This is an important contribution to the domains of chiral supramolecular aggregates and their complexes with (supra-)colloidal plasmonic systems, based on comprehensive joint experimental/simulation work. The site-dependent chiroptical properties of such complexes were studied in a systematic manner, and the analysis was thoroughly executed for the most part. In my view, this paper is potentially suitable for publication in Nature Communications, however, there is also the need to first put some more additional work in to gain a better physical understanding, especially of the satellite systems, and thus, for being able to fully assess the impact of this work.

Thus, the following points need to be addressed:

For the satellite complexes shown in Figure 4, the simulated optical properties do not align with the experimental system. This is true for g-factors and also the extinction spectra of the satellite clusters (for which calculated spectra are shown in Fig. S35). In the caption of Fig. S35, it is argued this discrepancy may be due to a dispersion with regard to the number of satellite particles per clusters. This is a testable prediction that should be substantiated by calculations for varying satellite numbers. In any case, the paper would benefit from openly discussing these discrepancies in the main text, rather than having them mentioned only in a Figure caption in the supporting material.

The second point is somewhat related. From the experimental side, the synthesized materials need to be more thoroughly structurally characterized and described. What is the size distribution for the different gold nanoparticle batches? At the moment, only average values are given. How were the differently sized gold nanoparticles synthesized? What is the structural identity of the satellite clusters? Can aggregation of these clusters be ruled out? I think especially for the satellite systems ensemble methods for structural characterization, like light scattering and small-angle X-ray scattering, are almost necessary to adequately study them.

Figure 1b: FRET-type quenching of fluorescence is discussed. Does the fluorescence quenching follow the associated d^{-6} distance dependence?

The main text reads rather descriptive at times, and it may improve readability to make some general conclusive statements at the ends of paragraphs/sections.

More information about the performed DFT calculations is necessary.

Incomplete sentence in lines 241-241.

Caption of Figure S25: Data in the main text is wrongly referenced. Why is smoothing performed at all when it needn't be?

Reviewer #2 (Remarks to the Author):

Major revisions

This manuscript presents a very interesting and detailed study of chiral plasmonic DNA oligo-metal nanoparticle complexes and how the intermolecular interactions controlling complex formation lead to the surface plasmons from the nanoparticle surfaces modifying the excited electronic states. The authors have employed a number of physical characterisation techniques in combination with DFT calculations of the electronic states, though the extensive use of circular dichroism (CD) spectroscopy presents the main source of novelty in this work. The experimental research appears extensive and to

have been performed reliably, and we agree with the analysis and conclusions drawn. We also are happy that this manuscript will interest the many researchers now working in chiral plasmonics and the broader plasmonic materials field. However, this wasn't an easy manuscript to read through as much of the text is too verbose, making the discussion difficult to follow at times while also leaving out some details that we believe are important for understanding the work reported. Therefore, we recommend that this manuscript be accepted for publication after the major revisions listed below are addressed.

Results and Discussion:

The text is too lengthy and verbose, and that it should be significantly reduced by abbreviating some of the material. For example, please condense the descriptions of Fig 1. Fig 1c could be supplementary information (SI), panel 1b also should be moved to SI.

The CD data contribute the main novelty in this work, the Fluorescence and Absorbance data are not novel and should be moved to SI. The discussion of the Fluorescence and Absorbance data in the main manuscript should then be correspondingly abbreviated.

Fig 1. Add markers to make the red shifting easier to see.

The chemical structures of the S-BINOL and BODIPY core moieties should be clarified.

Please clarify, by the 'free chromophore' mentioned in the text do the authors just mean the BDP moiety i.e., nt free? If so, why doesn't this appear in Fig 1b and c?

Sample identities need to be clearer, particularly with Fig 1d and Fig 1e. We assume that the 2nt sample includes the metal NP while the cBDP-2nt sample doesn't? If so, please clarify.

In the "Frenkel-CT mixed intermolecular couplings on nano-confined surface" section, the first paragraph is too long winded an explanation, and needs to be more succinct.

Fig 2. The meaning of the T6 and T30 labels as well as the "low", "high" and "middle" labels are all unclear and need to be defined.

In their discussion of the excitonic coupling for their cBDP dimers the authors state "Such CD splitting, similar to the observation in 7nt complexes (Fig. 2b), was obviously due to the intermolecular excitonic correlations." As Nature Communications targets a general audience, we doubt that many potential readers will find this obvious. This sentence should be rephrased, and a clearer explanation be provided.

The process of forming cBDP dimers with specific separations is unclear, more detail is required to explain how the dimers were produced and stabilised. What were the specific sequences of these DNA oligos? What was the pH used, and the final concentration of the dimer preparation? How long were they stored before use?

We are confused by the authors' use of the description "solar system assemblies" throughout the manuscript. While we can see the analogy of these constructs with our solar system's planetary structure, it is a confusing terminology that they are using here.

Many of the Figure panels are detailed but the font sizes are so small that they and their text labels are difficult to clearly see e.g., Figs 3 and 4. Please make these larger.

The ordering of panels in Figures 1, 3 and 4 is inconsistent. Please reformat the figures for consistency.

The large amount of description in the captions for Figures 3 and 4 makes it difficult to follow as the reader goes back and forth between the text and the figure. While the detailed descriptions of these

Figures are informative, it is difficult to read at times because the text doesn't flow. This section should be restructured so that it can be more easily understood by the reader.

The 40 pages of Supplementary Information seems excessive, with much of this material regarding characterisation of the nanoparticles. Only 4 of the Figures and one Table in Supplementary Information are referred to in text. We recommend that if these Figures were not significant enough to be referred to in the manuscript, they are probably not significant enough to be included in the SI document.

We liked the concept of this work and it's execution by the authors, and we hope that these comments can strengthen the manuscript leading to its wide appreciation by the plasmonic materials community.

Reviewer #3 (Remarks to the Author):

Reviewer #4 (Remarks to the Author):

The manuscript describes the combined experimental and theoretical studies on the modification of CD spectra of chiral chromophores on Au nanostructure, which is induced by molecular aggregation and by plasmon-molecule optical interactions.

The subject of the manuscript, the chiral induction of achiral plasmonic nanoparticle and chiral amplification by plasmonic nanoparticle, is important for fundamental nano-scale light-molecule interaction. Also, the subject has implications in chiral molecule sensing and chiro-optical device development. In this regard, the topic will be of great interest to a broad readership of Nat. Comm. There already exist numerous experimental and theoretical reports on this subject. I find this work an important step toward the understanding of the problem in the following aspects:

1. Chiral visible chromophore resonating with visible plasmon resonance: Previous experimental works exclusively study the off-resonant interactions of chiral UV-chromophore and plasmons, whereas the current work employs specially designed chiral visible chromophores that are fully resonant with visible plasmons. This resonant chiral induction has never been demonstrated before.
2. Molecular aggregation (excitonic and/or CT-complex) effect: Because of the reason "1", previous works could not address how the aggregation of chromophores affects the chiral interactions. The current work has nicely demonstrated such an effect for the first time.
3. Geometry control: Employing the DNA sequence programming, the authors tuned the NP-chromophore distance, which affects the chiral induction. This effect is not unexpected but has never been demonstrated before.

Despite a very nice set of data and interpretation, the manuscript contains missing information, somewhat sloppy terminology, and hard-to-read sentences (see below for detail). For the reasons above, I recommend the manuscript be published to Nat. Comm., provided that the authors could improve the structure/style/ writing of the manuscript, such that the text and figures effectively deliver the scientific findings.

Below, I list a few of such problems:

*Just for the completeness, the authors should provide CD spectra of metal-molecule complex with BODIPY-(R)-Binol.

*MISSING INFORMATION

Line 133: “..between a quencher and cBDP..”

The main text does not provide what kind of quencher molecules have been employed. In the caption of Fig. 1, it was specified that BHQ is the quencher. It still does not provide what exactly BHQ is.

Lines 242-246 “According to DFT calculations.... (Process II)”

The sentences are hard to understand. The corresponding figure (Fig 3i) and the associated caption do not fully deliver sufficient information.

Line 160 “...(denoted as rho in Fig. 2)..” and Figure 2e

How is rho defined or measured? What is the unit of rho in Figure 2e?

Figure 4c, x-axis, “distance”

What does the ‘distance’ refer to?

*SENTENCES THAT ARE HARD TO UNDERSTAND

Lines 202-205, “The decrease in CD ... correlated with the cBDP amounts”

Line 237: “...GSB bands was from the superposition of negative GSB and positive CT signals..”

*VAGUE STATEMENTS:

Line 213-214 “cBDP overcrowding could ... CD enhancement”

Line 225-226 “The CD internal ratio and absorption peak wavelength... dimensions”

Line 247: “multiplexed CT”

Response Letter

We appreciate the reviewers' time and effort in evaluating our work and providing valuable feedback. We have carefully checked through the revision comments and suggestions from the reviewers, and revised the manuscript and supplementary information accordingly. Our point-by-point responses are listed in the following.

Reviewer #1 (Remarks to the Author):

This is an important contribution to the domains of chiral supramolecular aggregates and their complexes with (supra-)colloidal plasmonic systems, based on comprehensive joint experimental/simulation work. The site-dependent chiroptical properties of such complexes were studied in a systematic manner, and the analysis was thoroughly executed for the most part. In my view, this paper is potentially suitable for publication in Nature Communications, however, there is also the need to first put some more additional work in to gain a better physical understanding, especially of the satellite systems, and thus, for being able to fully assess the impact of this work.

Response: We appreciate the reviewer for positive feedback and valuable suggestions. We have carried out more theoretical and experimental studies to provide clarity and improve the impact of this work.

Thus, the following points need to be addressed:

For the satellite complexes shown in Figure 4, the simulated optical properties do not align with the experimental system. This is true for g-factors and also the extinction spectra of the satellite clusters (for which calculated spectra are shown in Fig. S35). In the caption of Fig. S35, it is argued this discrepancy may be due to a dispersion with regard to the number of satellite particles per clusters. This is a testable prediction that should be substantiated by calculations for varying satellite numbers. In any case, the paper would benefit from openly discussing these discrepancies in the main text, rather than having them mentioned only in a Figure caption in the supporting material.

Response: Thank you for your comment. We agree that this hypothesis should be tested and substantiated by calculations for varying satellite numbers. We have revised the manuscript accordingly to provide a more comprehensive discussion. The additional simulations are shown below (Fig. R1 and Fig. R2). However, the focus of this paper is to reveal the essential roles of plasmon-affected intermolecular interactions between Frenkel-CT mixed excited states. Therefore, the detailed discussion of structural changes on the extinction spectra of core-satellites was only presented in the revised Supplementary Information as Fig. S45 and S46.

Fig. R1 Influence of structural changes on the optical responses of core-satellites. **a**, 3D model of core-satellites. **b**, Calculated extinction spectra of 40 nm particle with different dielectric constant of surroundings (ϵ_s). **c-e**, Calculated absorption (**c**), scattering (**d**) and extinction (**e**) spectra of 40 nm particle ($\epsilon_s = 1.77$) and core-satellites with different number of satellites per ring (the core-satellite gap is 10 nm, and the effective dielectric constant is set 2.00).

Fig. R2 Schematic of core-satellites (a), and calculated extinction spectra (b) of 40 nm particle ($\epsilon_s = 1.77$) and core-satellites with different number of satellites per ring (the core-satellites gap is 7.5 nm, and the effective dielectric constant is set 2.00).

The updated discussion in Supplementary Information is as follows.

We experimentally observed large extinction red-shift of core-satellite superstructures relative to the individual core particles (Fig. 4g). To study such spectral shift, a 3D model of the superstructure was built theoretically as depicted in Fig. S45a. In fact, there are many parameters that need to be considered to meet the experimental conditions. Firstly, we noticed that the dielectric constant of surroundings could be modified as a result of local densification of DNA strands after assembling (*Phys. Rev. B* 1972, 6, 4370-4379; *J. Chem. Phys.* 1983, 79, 6130-6139). Also, the number of satellite particles in the superstructures could be varied between one another. Furthermore, the core-satellite gap sizes can be affected by the flexibility of double-stranded DNA linkers. Thus, we systematically studied the influence of change in the dielectric constant of surroundings, the number of satellite particles and the core-

satellite gaps. As shown in Fig. S45b, increase of the dielectric constant of surroundings around the AuNPs causes clear red-shift of extinction spectra. More red-shift can be generated with increasing the number of satellites (Fig. S45c-e). Similarly, reducing the core-satellite gap, for example, from 10 nm to 7.5 nm, also leads to an obvious red-shift (Figs. S45e and S46b). Therefore, we concluded that the observed extinction red-shift of core-satellite superstructures was attributed to the core-satellite plasmonic couplings but affected by structural variations.

The second point is somewhat related. From the experimental side, the synthesized materials need to be more thoroughly structurally characterized and described. What is the size distribution for the different gold nanoparticle batches? At the moment, only average values are given. How were the differently sized gold nanoparticles synthesized? What is the structural identity of the satellite clusters? Can aggregation of these clusters be ruled out? I think especially for the satellite systems ensemble methods for structural characterization, like light scattering and small-angle X-ray scattering, are almost necessary to adequately study them.

Response: Thank you for the comments and questions. The size distribution for the different gold nanoparticle (AuNPs) batches and their synthesis methods have been carefully checked and added in the Supplementary Information.

Synthesis of AuNPs

3 nm AuNPs were synthesized by the reduction of tetrachloroauric acid (HAuCl_4) with sodium borohydride (NaBH_4) and sodium citrate ($\text{C}_6\text{H}_5\text{Na}_3\text{O}_7$) following published procedures (*Chem. Mater.* 2016, 28, 1066-1075). 5 nm AuNPs were synthesized using sodium citrate and tannic acid as mixed reducing agents following published procedures (*Eur. J. Cell Biol.* 1985, 38, 87-93). 13 nm and 40 nm AuNPs were synthesized by Fren's method and seeded growth process following published procedures (*Small* 2013, 13, 2308-2315), respectively.

The TEM images and size statistics of AuNPs are shown below (Fig. R3) and added in the revised manuscript as Fig. S1.

Fig. R3 TEM images (a-d) and size distribution (e-h) of 3 nm AuNPs (diameter = 3.1 ± 1.1 nm), 5 nm AuNPs (diameter = 5.5 ± 1.2 nm), 13 nm AuNPs (diameter = 13 ± 3 nm) and 40 nm AuNPs (diameter = 40 ± 7 nm).

As for the structural identity of the satellite clusters, we followed the reviewer's suggestion using dynamic light scattering (DLS) and small-angle X-ray scattering (SAXS) to further characterize the ensemble of satellite clusters. The DLS and SAXS characterization methods have been added in the revised manuscript:

Dynamic light scattering

Dynamic light scattering (DLS) was measured on a Zetasizer Nano ZS analyzer (Malvern Instruments, Malvern Ltd). All samples were directly measured after gel electrophoresis and purification. The average hydrodynamic diameter of different

samples was calculated.

Small-angle X-ray scattering

Small-angle X-ray scattering (SAXS) measurements were performed on beamline BL16B1 of the Shanghai Synchrotron Radiation Facility. The instrument was equipped with an X-ray source with a wavelength of 1.24 Å and an area detector consisting of 3 × 8 panels (Pilatus 2M). Two-dimensional SAXS data were calibrated using a silver behenate standard. Saxsgui v2.15.01 was used to generate the one-dimensional integrated scattering profiles of intensity I vs scattering vector q.

The DLS and SAXS data are shown below (Fig. R4-5) and added in the revised manuscript as Figs. S37 and 4f.

Fig. R4 Size distribution of the core-satellites and the core control obtained by dynamic light scattering (DLS). The average size of core-satellites less than twice of the core size indicated no observable cluster aggregations.

Fig. R5 Small angle X-ray scattering (SAXS) profiles (f) of solar system-like superstructured assemblies. The inset shows three-dimensional (3D) tomographic reconstruction of a single assembly. The core with a diameter of 40 nm has its first form factor minimum at a q -value of about $0.25 nm^{-1}$, whereas the satellites have theirs at $1.5 nm^{-1}$. The core-satellites curve showed resembling patterns with previous report (*ACS Nano* 2016, 10, 5740-5750).

Fig. R6 Agarose gel (a and b) and TEM images (c) of solar system-like superstructured assemblies. Left inset in (c) is a zoom-in image of a single superstructured assembly, scale bar is 20 nm. The single narrow band of target products in agarose gel demonstrated mono-dispersity of superstructured assemblies.

Also, we updated the agarose gel images in Fig. S34, as shown above. The sharp and single gel band of target products (*Angew. Chem.* 2016, 128, 14508-14512; *Small* 2016, 12, 4662-4668) along with SAXS and DLS data all proved monodispersed core-satellite clusters with good stability and narrow size distribution, basically ruling out aggregations.

Figure 1b: FRET-type quenching of fluorescence is discussed. Does the fluorescence quenching follow the associated d^{-6} distance dependence?

Response: Thank you for this comment. Site-specific dye conjugation with DNA has been well recognized (*Acc. Chem. Res.* 2017, 50, 1367-1374; *Proc. Natl. Acad. Sci. U. S. A.* 2004, 101, 5488-5493), which is also the advantage of DNA nanotechnology. The FRET-type quenching was discussed for confirming the site-specificity of cBDP-DNA conjugates. The FRET efficiency is theoretically given by the following equation:

$$E_{\text{FRET}} = \frac{R_0^6}{R_0^6 + R_{\text{DA}}^6} \quad (\text{Eq. R1})$$

where R_0 is the distance between donor and acceptor that gives 50% FRET efficiency and R_{DA} is the actual distance between donor and acceptor. Since R_0 is a constant for a

pair of given donor and acceptor (or quencher), Eq. R1 could be written as:

$$E_{\text{FRET}} = \frac{1}{1+(R_{\text{DA}}/R_0)^6} = \frac{1}{1+d^6} \quad (\text{Eq. R2})$$

R_0 could be estimated as 6-7 nm with web-based FRET calculator (*Angew. Chem.* 2022, 134, e202207; <https://www.fpbase.org/fret/>).

Fig. R7 Fluorescence spectra of cBDP-DNA monomer and cBDP-BHQ1 FRET pair with varying inter-distances.

E_{FRET} could be roughly defined as

$$E_{\text{FRET}} = 1 - \frac{\text{Fluorescence intensity of cBDP-BHQ1 FRET pair @ 510 nm}}{\text{Fluorescence intensity of cBDP-DNA monomer @ 510 nm}} \quad (\text{Eq. R3})$$

Then, $E_{\text{FRET}} = 0.25$ for 14bp pair (Fig. R7) and a theoretical curve (light green and orange lines in Fig. R8) could be established (*J. Am. Chem. Soc.* 2017, 139, 9271-9280).

Fig. R8 FRET efficiency (E_{FRET}) as a function of distance from the tethering point of cBDP chromophore to that of BHQ1 quencher in the double-stranded DNA. The green and yellow lines are theoretical curves based on the equation $E_{\text{FRET}} = \frac{1}{1+(R_{\text{DA}}/R_0)^6} =$

$$\frac{1}{1+d^6}$$

Fig. R9 Structural formula of a pair of cBDP and BHQ1 in DNA duplex.

Taking the 2bp pair as an example, its structural formula is shown in Fig. R9. The distance between cBDP chromophore and its tethering point at the end of DNA is about 1 nm, so the estimated R_{DA} is around 2.6 nm ($= [(1+0.68)^2+2^2]^{1/2}$ nm). The reasons for the deviation from the theoretical curve are complex, including possible changes in distance owing to tilting of the DNA, linker flexibility, and photophysical effects (*Proc. Natl. Acad. Sci. U. S. A.* 2008, 105, 18337-18342; *ChemPhysChem* 2011, 12, 689-695). In addition, the molecular orientation is another major parameter that affects the FRET efficiency. Both the distance and the orientation between cBDP and BHQ1 change when the number of separating bases is increased (*J. Am. Chem. Soc.* 2017, 139, 9271-9280; *Nat. Methods* 2018, 15, 669-676). Here, the pair of cBDP and BHQ1 roughly shows FRET-type quenching associated with their inter-distance.

The main text reads rather descriptive at times, and it may improve readability to make some general conclusive statements at the ends of paragraphs/sections.

Response: Thank you for your suggestions on the main text. We have significantly revised the corresponding paragraphs in the text (highlighted in yellow) and make conclusions at the end of paragraphs (highlighted in green). We believe such changes would make the manuscript more readable.

More information about the performed DFT calculations is necessary.

Response: The DFT calculations have been revised in **Methods** of main text:

All the calculations were performed with Gaussian16 in the framework of the density functional theory with the B3LYP method. The 6-311G(d,p) basis set is used for the geometry optimization (C, H, O, N and B atoms). The long-range van der Waals interaction is described by the DFT-D3 approach.⁷¹ The optimal structure was calculated using PCM solvent model in aqueous solution. Molecular structures were

visualized using GaussView 6.1.1. For a given geometry, the charge distribution in the ground state was derived from the Mulliken charges on the donor ((*S*)-BINOL moiety) and the acceptor (BODIPY core). In addition, we applied an electric field of magnitude 0.005 V/Å in the direction parallel or perpendicular to the molecular principal axis to analyze the electron transfer within the molecule.

Incomplete sentence in lines 241-241.

Response: Thank you for the careful review and pointing out this incompleteness from unexpected editing. We have corrected this sentence as following:

When the BODIPY core was spatially close to the (*S*)-BINOL moiety (dimer-3) with an appropriate relative orientation, intermolecular CT occurred and resulted in apparent change of charge density distribution (Fig. 3h).

Caption of Figure S25: Data in the main text is wrongly referenced. Why is smoothing performed at all when it needn't be?

Response: We appreciate your careful review and efforts. We have corrected the Caption of Figure S25 (updated as Figure S30).

Fig. S30 CD spectra of single-NP complexes with size variation. The CD intensity of all 2nt complexes is the highest, with little difference between 7nt and 14nt complexes for each AuNP size. Asymmetric CD lineshape can be observed in both 7nt complexes of 3 nm and 5 nm AuNP sizes. The data in main text Fig. 4b are experimental results (appropriate spectral smooth fitting for 13 nm and 40 nm complexes).

Actually, we only performed smooth fitting at 13 nm and 40 nm complexes (Fig. S33c,d), because the scattering effect of larger AuNPs has a great influence on the signal-to-noise ratio of CD spectra (*Nano Lett.* 2016, 16, 5962-5966; *Nanoscale*, 2014, 6, 2307-2315). We will provide all the raw spectral data in **Source Data**.

Reviewer #2 (Remarks to the Author):

Major revisions

This manuscript presents a very interesting and detailed study of chiral plasmonic DNA oligo-metal nanoparticle complexes and how the intermolecular interactions controlling complex formation lead to the surface plasmons from the nanoparticle surfaces modifying the excited electronic states. The authors have employed a number of physical characterisation techniques in combination with DFT calculations of the electronic states, though the extensive use of circular dichroism (CD) spectroscopy presents the main source of novelty in this work. The experimental research appears extensive and to have been performed reliably, and we agree with the analysis and conclusions drawn. We also are happy that this manuscript will interest the many researchers now working in chiral plasmonics and the broader plasmonic materials field. However, this wasn't an easy manuscript to read through as much of the text is too verbose, making the discussion difficult to follow at times while also leaving out some details that we believe are important for understanding the work reported. Therefore, we recommend that this manuscript be accepted for publication after the major revisions listed below are addressed.

Response: We are grateful for your recognition of the broad interest and impact of our study. We will make necessary changes to streamline the discussion and provide a more comprehensive understanding of this work. Specifically, we will focus on reducing unnecessary verbosity while maintaining clarity and coherence. We will also ensure that important details, particularly those related to the experimental procedures and results, are adequately described to facilitate reproducibility and understanding.

Results and Discussion:

The text is too lengthy and verbose, and that it should be significantly reduced by abbreviating some of the material. For example, please condense the descriptions of Fig 1. Fig 1c could be supplementary information (SI), panel 1b also should be moved to SI.

Response: Thank you for this suggestion. The discussion through the whole text has been significantly revised, largely reducing verbose descriptions. Fig. 1 has been updated with caption as following:

Fig. 1 Structures and optical properties of site-specific hybrid complexes. a, Schematic of site-specific hybrid complexes and structural formula of cBDP-DNA conjugates. **b,** CD spectra of Au complexes with cBDP (*S*) and cBDP (*R*). **c, d,** CD spectra of cBDP (*S*), Ag (**d**) and Au (**e**) complexes with cBDP (*S*). The cBDP surface concentrations in the three complexes were approximately the same in (**c**) and (**d**), respectively.

The CD data contribute the main novelty in this work, the Fluorescence and Absorbance data are not novel and should be moved to SI. The discussion of the Fluorescence and Absorbance data in the main manuscript should then be correspondingly abbreviated.

Response: As responded to the above comment, Fig. 1 and its related contents in the main manuscript have been revised accordingly.

Fig 1. Add markers to make the red shifting easier to see.

Response: We have added red arrows in Fig. 1c,d to indicate the red shifting.

The chemical structures of the *S*-BINOL and BODIPY core moieties should be clarified.

Response: Thank you for this constructive comment. We updated the structural formula of cBDP-DNA conjugates in Fig. 1a. BODIPY core is highlighted in blue and BINOL moiety is highlighted in red.

Please clarify, by the ‘free chromophore’ mentioned in the text do the authors just mean the BDP moiety i.e., nt free? If so, why doesn’t this appear in Fig 1b and c?

Response: We apologize to the confusion caused. ‘Free chromophore’ means the chiral dye, i.e., BODIPY-*S*-BINOL, without DNA. We have updated it with ‘pure cBDP molecules’.

Sample identities need to be clearer, particularly with Fig 1d and Fig 1e. We assume that the 2nt sample includes the metal NP while the cBDP-2nt sample doesn't? If so, please clarify.

Response: Thank you for your feedback and we apologize for any confusion caused. 2nt sample refers to the metal NP with cBDP molecule, while cBDP-2nt sample refers to cBDP-DNA conjugate, and we have updated cBDP-DNA in Fig. 1 caption and the main text.

Fig. 1 Structures and optical properties of site-specific hybrid complexes. a, Schematic of site-specific hybrid complexes and structural formula of cBDP-DNA conjugates. **b,** CD spectra of Au complexes with cBDP (*S*) and cBDP (*R*). **c, d,** CD spectra of cBDP (*S*), Ag (**d**) and Au (**e**) complexes with cBDP (*S*). The cBDP surface concentrations in the three complexes were approximately the same in (**c**) and (**d**), respectively.

In the “Frenkel-CT mixed intermolecular couplings on nano-confined surface” section, the first paragraph is too long winded an explanation, and needs to be more succinct.

Response: Thank you for your suggestion. The paragraph is rewritten as following:

The surface concentration (denoted as ρ in Fig. 2) and separation of cBDP molecules on nanoparticles were thoroughly tuned to analyze intermolecular couplings (Fig. 2a). The positive CD band of 7nt complexes reproducibly appeared asymmetric or even became split (Fig. 2b). Such split can be attributed to excitonic correlations between cBDP at close proximity on the nanoparticle surface.⁴⁵ The asymmetric split may arise from the overlapping of intrinsic positive CD band of cBDP with bisignate exciton-coupled CD. These two origins of chirality may also lead to asymmetry of CD lineshape rather than obvious splitting with a change of surface separation and concentration of cBDP. Comparatively, the CD lineshape of 2nt complexes, which presumably had higher excitonic coupling strength due to closer proximity, remained symmetric (Fig.

2c), indicating more complex CD origins. The CD lineshape of 14nt complexes was also symmetric mainly due to the intrinsic chirality of cBDP at large intermolecular distances (Figs. S22a and S30). Thus, the distance-dependent intermolecular couplings can be considered to cause these CD lineshape differences.

Fig 2. The meaning of the T6 and T30 labels as well as the “low”, “high” and “middle” labels are all unclear and need to be defined.

Response: Thank you for the suggestion. We revised the caption and provided a clear explanation of these labels. T₆ and T₃₀ stands for DNA spacer co-modified on nanoparticles along with cBDP-DNA. Their detailed DNA sequence is in **List of DNA strands** in Supplementary Information. Additionally, we defined the “low”, “high”, and “middle” labels to ensure their meaning is properly conveyed.

Fig. 2 Influence of intermolecular interactions within cBDP ensemble. a, Schematic of tuning of surface separation and concentration of cBDP. T₆ and T₃₀ stands for DNA spacer co-modified on nanoparticles. b-d, CD spectra of 7nt (b) and 2nt (c) complexes, and g-factors (d) of 14nt, 7nt, and 2nt complexes with varying surface separations and concentrations of cBDP. e, CD enhancement factor and internal ratio of 2nt complexes with varying surface concentrations of cBDP. “Low” refers to low cBDP/Au nanoparticle molar ratio during synthesis, i.e., n_{cBDP}: n_{Au} = 25: 1, “middle” refers to n_{cBDP}: n_{Au} = 50: 1, and “high” refers to n_{cBDP}: n_{Au} = 100: 1 (see Table S1-S3 for details). Co-modification of Au nanoparticles with T₃₀ resulted in the lowest molar ratio.

In their discussion of the excitonic coupling for their cBDP dimers the authors state “Such CD splitting, similar to the observation in 7nt complexes (Fig. 2b), was obviously due to the intermolecular excitonic correlations.” As Nature Communications targets a general audience, we doubt that many potential readers will find this obvious. This sentence should be rephrased, and a clearer explanation be provided.

Response: Thank you for the insightful comment. The rephrased sentence reads: Such CD splitting, a phenomenon of Davydov splitting (*ACS Nano* 2022, 16, 1301-1307; *J. Phys. Chem. B* 2020, 124, 8042-8049), similar to that in 7nt complexes (Fig. 2b), arises from coherent excitonic correlations between cBDP molecules.

Davydov splitting is largely reported in molecular excitonic couplings. We cited relevant references for reader’s understanding.

The process of forming cBDP dimers with specific separations is unclear, more detail is required to explain how the dimers were produced and stabilised. What were the specific sequences of these DNA oligos? What was the pH used, and the final concentration of the dimer preparation? How long were they stored before use?

Response: Thank you for the comments and questions. Two single-stranded DNA that

satisfy Watson-Crick base-pairing rule (Adenines pairing with Thymines, and Guanines pairing with Cytosines) can form a DNA duplex (*Angew. Chem. Int. Ed.* 2014, 53, 11366-11369; *J. Phys. Chem. B* 2016, 120, 4009-4018; *Proc. Natl. Acad. Sci. U. S. A.* 1986, 83, 3746-3750), by annealing in alkaline buffer at a certain NaCl concentration. The DNA duplex takes the form of a “twisted ladder” structure primarily held together by hydrogen bonding, as exemplified by the dimer iii below:

Fig. R10 (a) Schematic of DNA templating of cBDP dimer iii and (b) its structural formula. The note “2nt” in strand 3 means that cBDP is conjugated to the 2nd nucleobase position from the 3’ terminus, and “1nt” in strand 4 means unpaired base number after hybridization between strand 3 and 4.

Here is additional information to address your questions:

What were the specific sequences of these DNA oligos?

DNA sequences for cBDP dimers

DNA strands	Sequence
1	TTTTTTTTTTTTTTTTT/BODIPY-(S)-BINOL/TTTTTTTTTTTTTT-HSSH
2	TTTTTTTTTTTTTTTTTTTTTTTTTTTTTTTTT/BODIPY-(S)-BINOL/TTTTTT-HSSH
3	TTTTTTTTTTTTTTTTTTTTTTTTTTTTTTTTT/BODIPY-(S)-BINOL/T-HSSH
4	BODIPY-(S)-BINOL/TAAAAAAAAAAAAAAAAAAAAAAAAAAAAAAAAAAAAA
5	BODIPY-(S)-BINOL/TTAAAAAAAAAAAAAAAAAAAAAAAAAAAAAAAAAAAA
6	BODIPY-(S)-BINOL/TTTTAAAAAAAAAAAAAAAAAAAAAAAAAAAAAAAAAAAA

DNA strands 1-3 are cBDP-DNA conjugates, namely, cBDP-DNA (14nt), cBDP-DNA (7nt) and cBDP-DNA (2nt) respectively. The notes 14nt, 7nt and 2nt in strands 1-3 mean the 14th, 7th and 2nd thymine from the 3’ end. DNA strands 4-6 are also cBDP-DNA conjugates, which are used to assemble different cBDP dimers with DNA strands 1-3, as shown in main text Fig. 3a.

What was the pH used, and the final concentration of the dimer preparation?

The pH was 8.0 and the final concentration of the dimer was 2.5 μM . We revised the **Method** accordingly:

Synthesis of cBDP molecular dimers

Two complementary DNA strands conjugated with cBDP were added together in equal molar amounts into 0.5 \times TE buffer (5 mM Tris, 0.05 mM EDTA, pH = 8.0) with 300 mM NaCl. The sample was annealed by cooling from 60 $^{\circ}\text{C}$ to 30 $^{\circ}\text{C}$ (1 $^{\circ}\text{C}/20\text{min}$) with a final concentration of 2.5 μM , then used directly for further characterizations.

How long were they stored before use?

As responded above, they were used directly for further characterizations after annealing at the same day.

We are confused by the authors' use of the description "solar system assemblies" throughout the manuscript. While we can see the analogy of these constructs with our solar system's planetary structure, it is a confusing terminology that they are using here.

Response: Thank you for pointing out this confusion. Plasmonic core-satellite superstructures were first reported in 1998 through DNA-directed assembly (*J. Am. Chem. Soc.* 1998, 120, 12674-12675). Primarily, core-satellite superstructures consist of a single core or multiple cores surrounded by smaller peripheral particles (*Chem. Rev.* 2019, 119, 12208-12278; *J. Am. Chem. Soc.* 2012, 134, 12083-12090; *Adv. Optical Mater.* 2014, 2, 65-73; *Nano Lett.* 2012, 12, 2645-2651; *ACS Nano* 2012, 6, 7199-7208). It must be noted that the core-satellite superstructure complexes reported here contain not only AuNP core and satellites, but also cBDP molecules specifically arranged surrounding the small AuNP satellites. One could imagine the large AuNP as the sun, the small AuNPs as the planets, and the cBDP molecules as the satellites. This presents an obviously different vision from the previous report (*J. Am. Chem. Soc.* 2012, 134, 12083-12090; *Adv. Optical Mater.* 2014, 2, 65-73; *Nano Lett.* 2012, 12, 2645-2651; *ACS Nano* 2012, 6, 7199-7208). We hope to give the general audience an intuitive picture. But, to avoid confusion, we rephrased the terminology as "solar system-like assemblies" when we really need to emphasize the hybrid nature of the complexes. While, we still used the "core" and "satellite" terminology when referring to the AuNPs in the superstructures.

Many of the Figure panels are detailed but the font sizes are so small that they and their text labels are difficult to clearly see e.g., Figs 3 and 4. Please make these larger.

The ordering of panels in Figures 1, 3 and 4 is inconsistent. Please reformat the figures for consistency.

Response: Thank you for pointing out these details. Basically, the figures were

presented following *Nature Communications'* guideline. We have organized Fig. 4 in a manner that maintains the consistency for readers' convenience while considering the logic and aesthetics of the images. As a matter of fact, we also did try to reformat Fig. 3, but find it hard to follow the normal ordering, such as from left to right, to give a satisfied presentation. Noticeably, *Nature Communications* have published articles containing inconsistent figures when comparing relevant data (*Nat. Commun.* 2022, 13, 7841; *Nat. Commun.* 2023, 14, 8137).

The large amount of description in the captions for Figures 3 and 4 makes it difficult to follow as the reader goes back and forth between the text and the figure. While the detailed descriptions of these Figures are informative, it is difficult to read at times because the text doesn't flow. This section should be restructured so that it can be more easily understood by the reader.

Response: We apologize for the difficulty caused. Due to the limit of data presented in the main text, it is difficult to separate these figures. Therefore, we restructured these sections by adjusting the position of the relevant figures to make them more closely related to the main text. Moreover, we have moved some details from the caption to the main text and added necessary details correspondingly, with yellow highlight, to enhance the readability of the text.

The 40 pages of Supplementary Information seems excessive, with much of this material regarding characterisation of the nanoparticles. Only 4 of the Figures and one Table in Supplementary Information are referred to in text. We recommend that if these Figures were not significant enough to be referred to in the manuscript, they are probably not significant enough to be included in the SI document.

Response: Thanks for the comments. We acknowledge that the SI contains a significant amount of material related to the characterization of not only the nanoparticles, but also cBDP molecule, cBDP-DNA conjugates and Au complexes (mostly depicted by Au spheres in SI figures). We understand your point that if certain figures were not significant enough to be referred to in the main manuscript, they might not be essential for inclusion in the SI document. However, we would like to mention that the characterization of materials is an important aspect of our study, as it provides crucial information about their physical and chemical properties. Reviewer 1 also pointed out that 'the synthesized materials need to be more thoroughly structurally characterized and described'. While we agree that not all details may be necessary for the main manuscript, we believe that including this information in the SI allows interested readers to delve deeper into the technical aspects of our work.

In light of your feedback, we will carefully reassess the content of the SI, ensuring that only the most relevant and essential figures and tables are included. We will also make

sure that any references to the SI in the main manuscript are appropriate and necessary for the readers' understanding.

We liked the concept of this work and it's execution by the authors, and we hope that these comments can strengthen the manuscript leading to its wide appreciation by the plasmonic materials community.

Response: Thanks again for the supportive evaluation on our work. These insightful comments will undoubtedly help strengthen the quality of the work.

Reviewer #3 (Remarks to the Author):

Response: Thank the reviewer for the time and the valuable feedback. We have revised the manuscript accordingly.

Reviewer #4 (Remarks to the Author):

The manuscript describes the combined experimental and theoretical studies on the modification of CD spectra of chiral chromophores on Au nanostructure, which is induced by molecular aggregation and by plasmon-molecule optical interactions.

The subject of the manuscript, the chiral induction of achiral plasmonic nanoparticle and chiral amplification by plasmonic nanoparticle, is important for fundamental nano-scale light-molecule interaction. Also, the subject has implications in chiral molecule sensing and chiro-optical device development. In this regard, the topic will be of great interest to a broad readership of Nat. Comm.

There already exist numerous experimental and theoretical reports on this subject. I find this work an important step toward the understanding of the problem in the following aspects:

1. Chiral visible chromophore resonating with visible plasmon resonance: Previous experimental works exclusively study the off-resonant interactions of chiral UV-chromophore and plasmons, whereas the current work employs specially designed chiral visible chromophores that are fully resonant with visible plasmons. This resonant chiral induction has never been demonstrated before.
2. Molecular aggregation (excitonic and/or CT-complex) effect: Because of the reason "1", previous works could not address how the aggregation of chromophores affects the chiral interactions. The current work has nicely demonstrated such an effect for the first time.
3. Geometry control: Employing the DNA sequence programming, the authors tuned the NP-chromophore distance, which affects the chiral induction. This effect is not unexpected but has never been demonstrated before.

Despite a very nice set of data and interpretation, the manuscript contains missing information, somewhat sloppy terminology, and hard-to-read sentences (see below for detail). For the reasons above, I recommend the manuscript be published to Nat. Comm., provided that the authors could improve the structure/style/ writing of the manuscript, such that the text and figures effectively deliver the scientific findings.

Response: Thank you so much. We really appreciate your recognition of the novelty and importance of this work, especially by listing detailed supportive evidences. We also thank you for the constructive suggestions given. We have significantly revised the manuscript as highlighted.

Below, I list a few of such problems:

*Just for the completeness, the authors should provide CD spectra of metal-molecule complex with BODIPY-(R)-Binol.

Response: Thanks for this insightful comment. We synthesized BODIPY-(R)-BINOL following this suggestion. The CD spectrum of new metal-molecule complex with

BODIPY-(*R*)-BINOL (denoted as 2nt (*R*)) has been updated in Fig. 1b. Here, we only used 2nt (*S*) and 2nt (*R*) of 5 nm AuNPs for demonstration. We don't think it is necessary to re-compare all the complexes with systematic tuning of size and composition of NPs and site of cBDP, which could take years to complete.

Fig. 1 Structures and optical properties of site-specific hybrid complexes. a, Schematic of site-specific hybrid complexes and structural formula of cBDP-DNA conjugates. **b,** CD spectra of Au complexes with cBDP (*S*) and cBDP (*R*). **c, d,** CD spectra of cBDP (*S*), Ag (**d**) and Au (**e**) complexes with cBDP (*S*). The cBDP surface concentrations in the three complexes were approximately the same in (**c**) and (**d**), respectively.

*MISSING INFORMATION

Line 133: “..between a quencher and cBDP...”

The main text does not provide what kind of quencher molecules have been employed. In the caption of Fig. 1, it was specified that BHQ is the quencher. It still does not provide what exactly BHQ is.

Response: Thank you for pointing out this missing information. The quencher is a commonly used BHQ1 molecule and its structural formula has been shown below:

Fig. R9 Structural formula of a pair of cBDP and BHQ1 in DNA duplex.

We moved the following figure to Supplementary Information as Reviewer 2 suggested, along with the above structural formular as Figs. S14 and S15.

Fig. R7 Fluorescence spectra of cBDP-DNA monomer and cBDP-BHQ1 FRET pair with varying inter-distances.

Lines 242-246 “According to DFT calculations.... (Process II)”

The sentences are hard to understand. The corresponding figure (Fig 3i) and the associated caption do not fully deliver sufficient information.

Response: We apologize for any confusion caused. Currently, research mainly focuses on the intramolecular charge transfer (CT) of cBDP molecule (*Chem. Eur. J.* 2020, 26, 601-605; *J. Phys. Chem. B* 2023, 127, 45, 9781-9787), with very few reports on DFT calculations regarding intermolecular CT. In the case of the cBDP monomer, photoexcitation results in the formation of a CT state ($\text{BODIPY}^{\delta-}-(S)\text{-BINOL}^{\delta+}$). DFT calculations indicate that this CT state arises from electron transition from the HOMO (mainly located at the (S)-BINOL moiety) to the generated low-lying semi-vacant HOMO-1 (mainly located at the BODIPY core) (*Chem. Eur. J.* 2020, 26, 601-605; *J. Phys. Chem. B* 2023, 127, 45, 9781-9787). Based on the transient absorption spectra of dimer iii, we propose the possibility of intermolecular CT occurring between neighboring cBDP molecules during the photoexcitation process. DFT calculations support this speculation, indicating the potential for CT from the HOMO (mainly located at the (S)-BINOL moiety) of one cBDP molecule to the HOMO-1 (mainly located at the BODIPY core) of another cBDP molecule in the dimer, given the appropriate spatial conformation. We have cited these references (*Chem. Eur. J.* 2020, 26, 601-605; *J. Phys. Chem. B* 2023, 127, 9781-9787) in the sentences for readers' comprehension.

Line 160 “...(denoted as rho in Fig. 2)...” and Figure 2e

How is rho defined or measured? What is the unit of rho in Figure 2e?

Response: ρ is defined as the number of cBDP-DNA per AuNP, i.e., surface concentration of cBDP, and is obtained by $\rho = c_2/c_1$, where c_2 is the cBDP-DNA concentration and c_1 is the AuNP concentration. As exemplified in Fig. S17 (updated as S22), we used DTT to replace and separate the cBDP-DNA from AuNPs for quantification of ρ . The differences in absorption intensity near 520 nm before and after addition of DTT plus centrifugation can be used to estimate c_1 , and the CD intensity at 504 nm after addition of DTT and centrifugation can be used to estimate c_2 . Thus, ρ is obtained according to the definition without unit.

Fig. S22 CD spectra of 5 nm Au complexes before (solid lines) and after (dashed lines) DTT addition (a) and their corresponding absorption spectra (b). The blue shift and intensity decrease of the dashed lines in the CD and absorption spectra are related to the substitution of cBDP-DNA conjugates on the surface of AuNPs by DTT. It is obvious that after DTT addition, the cBDP-DNA conjugates were released to solution.

Figure 4c, x-axis, “distance”

What does the ‘distance’ refer to?

Response: The “distance” refers to the calculated distance to AuNP surface, as depicted in Fig. R11 (updated as Fig. 4d), i.e. “surface distance” in caption of Figure S32 (updated as S40).

Fig. R11 Electric field intensity as a function of surface distance for differently-sized nanoparticles.

Fig. S40 Electric field distribution maps for different sizes of AuNP (a) 3 nm, (b) 5 nm, (c) 13 nm, (d) 40 nm. The smaller the particle size, the more inhomogeneous the distribution of electric field intensity near the AuNP surface, for example, within a surface distance less than 5 nm.

*SENTENCES THAT ARE HARD TO UNDERSTAND

Lines 202-205, “The decrease in CD ... correlated with the cBDP amounts”

Response: The CD of Au complexes with a single cBDP molecule could be expressed

as

$$CD_{\text{cBDP-AuNP}} = a \cdot \text{Im}[(\hat{P} \cdot \vec{\mu}_{12}) \cdot \vec{m}_{21}] + b \cdot F(\vec{\mu}_{12}, \vec{m}_{21})$$

where $\vec{\mu}_{12}$ and \vec{m}_{21} are the electric and magnetic dipole moments of a single cBDP molecule, a and b are the coefficients which depend on the geometry, material system, and incident light frequency, ω . The first term originates from intramolecular dissipation and now includes the electric-field enhancement matrix \hat{P} . The second term is due to chiral dissipative currents in the AuNP induced by the electric dipole of cBDP molecule (*Nano Lett.* 2010, 10, 1374-1382). Therefore, the second term for Au complexes with cBDP molecular ensembles could be larger than that with only a single cBDP molecule (*Anal. Chem.* 2015, 87, 6455-6459), that is, the induced CD intensity was in-principle positively correlated with the cBDP amounts (*J. Am. Chem. Soc.* 2019, 141, 19336-19341; *Nano Lett.* 2013, 13, 3145-3151; *Angew. Chem. Int. Ed.* 2018, 57, 16452-16457; *Angew. Chem. Int. Ed.* 2022, 61, e202210730). We have added these references (*Nano Lett.* 2010, 10, 1374-1382; *Anal. Chem.* 2015, 87, 6455-6459; *J. Am. Chem. Soc.* 2019, 141, 19336-19341; *Nano Lett.* 2013, 13, 3145-3151; *Angew. Chem. Int. Ed.* 2018, 57, 16452-16457; *Angew. Chem. Int. Ed.* 2022, 61, e202210730) for readers' comprehension.

Line 237: "...GSB bands was from the superposition of negative GSB and positive CT signals..."

Response: We apologize for the obscure description. We revised the discussion after updating Fig. 3d-g with color image plots of the full-timescale transient absorption for readers' comprehensive views. Also, the transient absorption intensity at 514 nm is used to plot photogenerated electron kinetics, as shown in Fig. S27 and Table S4. We have revised discussion accordingly:

Femtosecond transient absorption (TA) spectroscopy was employed to examine the intermolecular CT in cBDP dimers, as shown in Fig. 3d-g. In the color image, the bright green color corresponded to the ground state bleaching (GSB) band at ~504 nm. The cBDP exhibited a negative GSB band that is associated with the $S_0 \rightarrow S_1$ transition⁵⁸. The GSB decay of dimer i (Fig. 3e) were quite similar to those of monomer, consistent with the above CD analysis. We performed kinetic analysis with exponential fittings of the decay curves at 514 nm to avoid the interference of laser incidence at 504 nm (Fig. S27). It was found out that dimer ii and iii proved intermediate states with a timescale of ~4 ps (Table S4), in contrast to dimer i and monomer. Such a difference indicated the existence of intermediate excited states, likely caused by intermolecular CT in closely-spaced cBDP dimers⁵⁸⁻⁶².

Fig. 3 Construction and intermolecular interactions of cBDP dimers. **a**, Schematic of DNA templating of cBDP dimers. The blue curved arrows depict the explored space of corresponding cBDP-DNA monomer. **b**, **c**, CD spectral evolution (**b**), maximum absorption wavelength and CD internal ratio of cBDP dimers (**c**). **d-g**, Color image plots of TA experiments of cBDP-DNA monomer (**d**) and dimers i-iii (**e-g**) at the same timescale. **h**, DFT calculated charge density distribution map of cBDP dimer with three different spatial orientations (Fig. S38). The red arrow in dimer-3 shows apparent intermolecular CT. **i**, Interference of surface localized plasmonic field with intra- and intermolecular CT via altering electronic couplings during electron migration.

Fig. R12 Transient kinetics for cBDP-DNA monomer (A) and dimers i-iii (B-D) at the same timescale (probed at 514 nm). The fits are fitted curves to the triple (A and B) and quadruple (C and D) exponential decay with the instrument response function.

Table S4. Fitted lifetime constants for cBDP-DNA monomer and dimers i-iii

	A@514 nm	B@514 nm	C@514 nm	D@514 nm
τ_1 (ps)	1.8	2.36	0.36	0.25
τ_2 (ps)	\	\	4.8	3.5
τ_3 (ps)	36	20	81	93
τ_4 (ps)	3272	1666	2884	3914

***VAGUE STATEMENTS:**

Response: Thank you for pointing out these vague statements, which help improve the clarity of the manuscript.

Line 213-214 “cBDP overcrowding could ... CD enhancement”

Response: The CD intensity of molecular ensembles is correlated with the cBDP amounts on AuNP surfaces. The excitonic couplings and charge transfer between cBDP are not only sensitive to their inter-distances but also their relative spatial orientations (*J. Phys. Chem. B* 2012, 116, 6751-6763; *J. Am. Chem. Soc.* 2006, 128, 16365-16372; *J. Phys. Chem.* 1972, 76, 2982-2986; *J. Am. Chem. Soc.* 2005, 127, 14208-14222). The crowding environment on AuNP surfaces could influence these parameters at the same

time, resulting in changes in the CD spectral lineshape or intensity (*Nature* 2005, 437, 1337-1340; *Angew. Chem. Int. Ed.* 2022, 61, e202210730). We have cited these references (*J. Phys. Chem. B* 2012, 116, 6751-6763; *J. Am. Chem. Soc.* 2006, 128, 16365-16372; *J. Phys. Chem.* 1972, 76, 2982-2986; *J. Am. Chem. Soc.* 2005, 127, 14208-14222; *Nature* 2005, 437, 1337-1340; *Angew. Chem. Int. Ed.* 2022, 61, e202210730) in the sentences for readers' comprehension.

Line 225-226 "The CD internal ratio and absorption peak wavelength... dimensions"

Response: This statement has been revised as:

The CD internal ratio variation and absorption peak shift well reflected the change of intermolecular distance and conformational dimension, which were closely related to Davydov splitting.

Line 247: "multiplexed CT"

Response: This statement has been changed to "multiple CT among cBDP ensemble".

Reviewer #1 (Remarks to the Author):

The authors convincingly addressed my questions and those of other reviewers in my view. Publication can now be fully recommended.

Reviewer #2 (Remarks to the Author):

We have reviewed the revised version of the manuscript and are happy with the authors' changes and their detailed responses to our recommendations. We are now happy to recommend publication of this revised manuscript and we hope that their paper receives broad attention and appreciation.

Reviewer #3 (Remarks to the Author):

Reviewer #4 (Remarks to the Author):

The revised manuscript fully addresses the comments and concerns. I recommend the manuscript to be published to the Nat. Comm. without further revision.